



# Emissions from village cookstoves in Haryana, India and their potential impacts on air quality

Lauren T. Fleming,[1] Robert Weltman,[2] Ankit Yadav,[3] Rufus D. Edwards,[2] Narendra K. Arora,[3] Ajay Pillarisetti,[4] Simone Meinardi,[1] Kirk R. Smith,[4] Donald R. Blake,[1] and Sergey A. Nizkorodov[1,*]

[1]Department of Chemistry, University of California, Irvine, CA 92697, USA
[2]Department of Epidemiology, University of California, Irvine, CA 92697, USA
[3]The Inclen Trust, Okhla Industrial Area, Phase-I, New Delhi-110020, India
[4]School of Public Health, University of California, Berkeley, CA 94720, USA

*Correspondence to:* Sergey Nizkorodov (nizkorod@uci.edu)

**Abstract.** Air quality in rural India is impacted by residential cooking and heating with biomass fuels. In this study, emissions of $CO$, $CO_2$, and 76 volatile organic compounds (VOCs) and fine particulate matter ($PM_{2.5}$) were quantified to better understand the relationship between cook fire emissions and ambient ozone and secondary organic aerosol formation. Cooking was carried out by a local cook and traditional dishes were prepared on locally built *chulha* or *angithi* cookstoves using brushwood or dung fuels. Cook fire emissions were collected throughout the cooking event in a Kynar bag (VOCs) and on PTFE filters ($PM_{2.5}$). Gas samples were transferred from a Kynar bag to previously evacuated stainless steel canisters and analyzed using gas chromatography coupled to flame ionization, electron capture, and mass spectrometry detectors. Filter samples were weighed to calculate $PM_{2.5}$ emission factors. Dung fuels and *angithi* cookstoves resulted in significantly higher emissions of most VOCs (p< 0.05). Utilizing dung-*angithi* cook fires resulted in twice as much of the measured VOCs compared to dung-*chulha*, and four times as much as brushwood-*chulha* with 84.0, 43.2, and 17.2 g VOC/ kg fuel carbon, respectively. This matches expectations, as the use of dung fuels and *angithi* cookstoves results in lower modified combustion efficiencies compared to brushwood fuels and *chulha* cookstoves. Alkynes and benzene were exceptions and had significantly higher emissions when cooking using a *chulha* as opposed to an *angithi* with dung fuel (benzene EFs: dung-*chulha* 3.18 g/ kg fuel carbon and dung-*angithi* 2.38 g/ kg fuel carbon). This study estimated that up to three times as much ozone and secondary organic aerosol may be produced from dung-*chulha* as opposed to brushwood-*chulha* cook fires. While aromatic compounds dominated as secondary organic aerosol precursors from all types of cook fires, benzene was responsible for the majority of SOA formation potential from all *chulha* cook fire VOCs, while substituted aromatics were more important for dung-*angithi*. Future studies should investigate benzene exposures from different stove and fuel combinations and model SOA formation from cook fire VOCs to verify public health and air quality impacts from cook fires.

## 1 Introduction

Parts of rural India are comprised of densely populated villages with ambient ozone and $PM_{2.5}$ levels that affect air quality for inhabitants (Bisht et al., 2015; Ojha et al., 2012; Reddy, 2012). For example, in the rural area of Anantapur in Southern India, monthly mean ozone levels varied between 29 ppbv in August during the monsoon season and 56 ppbv in April (Reddy, 2012). In Pantnagar, a semi-urban city, the maximum observed ozone concentration was 105 ppbv for one day in May, with the lowest average monthly maximum of 50 ppbv being in January (Ojha et al., 2012). In terms of $PM_{2.5}$ levels, Bisht et al., 2015 observed an average of 50 µg m⁻³ of $PM_{2.5}$ over July-November 2011 in rural Mahabubnagar. While measurements of $O_3$ and $PM_{2.5}$ in rural





India are relatively scarce, it has become clear that household combustion is a major contributor to ambient levels of these pollutants. For example Balakrishnan et al., 2013 measured $PM_{2.5}$ concentrations in households and observed 24-hour concentrations of 163 to 609 µg m$^{-3}$. Over the last half-decade, several researchers have, through independent studies, come to the conclusion that a significant fraction (22-52%) of ambient $PM_{2.5}$ is directly emitted from residential cooking and heating (Butt et al., 2016; Chafe et al., 2014; Conibear et al., 2018; GBD MAPS Working Group, 2018; Guttikunda et al., 2016; Klimont et al., 2017; Lelieveld et al., 2015; Silva et al., 2016).

Residences in India consume roughly 220, 86.5, and 93.0 Tg per year of dry matter of wood fuel, agricultural residues, and dung, respectively (Yevich and Logan, 2003). While the fraction of Indians using biomass cook fuels is decreasing, the total population is increasing such that biomass fuels are still being utilized at approximately the same overall level (Pandey et al., 2014). Emissions of primary $PM_{2.5}$ from residential cooking were estimated to be 2.6 Tg in India per year based on a compiled emissions inventory (Pandey et al., 2014). Additionally, Pandey et al., 2014 estimates that 4.9 Tg of non-methane volatile organic compounds (NMVOCs) are produced annually in India from residential cooking. This suggests that additional $PM_{2.5}$ mass may be formed via secondary pathways from the oxidation of NMVOCs and either nucleation of new particles or condensation onto existing $PM_{2.5}$. Alternatvely, these non-methane VOCs could contribute to photochemical ozone production in the presence of $NO_x$ (Finlayson-Pitts and Pitts, 2000).

In this study, we quantify emissions of CO, $CO_2$, and 76 different VOCs from 55 cook fires carried out by a local cook in a village home cooking typical meals. It is thus a substantially updated version of the work done in a simulated village houses in India and China in the 1990s, where 58 fuel-stove combinations were measured in semi-controlled conditions using water boiling tests including a number of non-biomass stoves although a similar set of pollutants were measured (Smith et al., 2000a, 2000b; Tsai et al., 2003; Zhang et al., 2000). This time, we measured emissions in field conditions from two traditional, locally-made cookstoves, the *chulha* and the *angithi*, the latter of which is primarily used for cooking animal fodder and simmering milk. We measured emissions from cookstoves with two kinds of biomass; the most popular biomass type brushwood (Census of India, 2011), and dung cakes. Our first objective is to characterize emissions of select VOCs and $PM_{2.5}$ from these fuel-stove combinations. Subsequently, with the aid of secondary organic aerosol (SOA) potential values from Derwent et al., 2010, incremental reactivity values from Carter, 1994, and second-order rate coefficients with OH combined with our emission factors, we estimate secondary organic aerosol forming potentials, ozone formed in a VOC-limited regime, and OH reactivity, respectively. Given their widespread use in India, emissions from these biomass-burning stoves are estimated to impact regional air quality due to both primary and secondary organic aerosol and ozone formation.

## 2 Experimental Methods

### 2.1 Field site and sample collection

The field office was located at the SOMAARTH Demographic, Development, and Environmental Surveillance Site in Palwal District, Haryana, India run by the International Clinical Epidemiological Network (INCLEN). The site consists of 51 villages in the area with roughly 200,000 inhabitants (Balakrishnan et al., 2015; Mukhopadhyay et al., 2012; Pillarisetti et al., 2014).

Samples were collected from cookstove emissions between August 5 and September 3, 2015. Cooking events occurred at a village kitchen in Khatela, Palwal District. A local cook was hired to prepare meals consisting of either *chapatti* or rice with vegetables using a *chulha* stove. The *angithi* stove burned only dung and was utilized to cook animal fodder, as is common in the area. Dung patties and brushwood were used in *chulha* stoves, and for the 13 mixed fuel cooking events dung and brushwood





were combined in a ratio determined by the cook's preference. Stoves and food ingredients were produced and fuels procured by the household or village. Additional information regarding the cooking events and set-up can be found in Fleming et al., 2018.

The sampling scheme is illustrated in Figure 1. Air sampling pumps (PCXR-8, SKC Inc.) created a flow of emissions through our sampling apparatus. Emissions were captured with three-pronged probes that were fixed 60 cm above the cookstove. $PM_{2.5}$

emissions and gases were sampled through cyclones (2.5 µm, URG Corporation). The resulting flow of particles was captured on either quartz or PTFE filters, while gases were collected in an 80 L Kynar bag throughout the entire cooking event. Flows were measured both before and after sampling to ensure they did not change by more than 10% using a mass flowmeter (TSI 4140). After the cooking event, pumps were turned off and a whole air sampler, consisting of a stainless steel canister (2L) welded to a Bellows-sealed valve (Swagelok), was filled to ambient pressure from the Kynar bag. Whole air samplers were flushed and

evacuated in the Rowland-Blake laboratory before being shipped to India. At the end of the measurement campaign, whole air samplers were shipped back to the laboratory and analyzed within two months of the end of the field measurements. A "grab" whole air sample was collected before cooking commenced each day. This served as a background for all cooking events sampled on that day.

2.2 Gas chromatography analysis

Colman et al., 2001 describes the VOC analysis protocol in detail. Briefly, a known amount of the whole air sample (WAS) flowed over glass beads cooled by liquid nitrogen. The flow was regulated by a mass flow controller and resulted in a 600 Torr drop in the pressure in the whole air sampler. High volatility gases such as $O_2$ and $N_2$ passed over the beads, while lower volatility gases adsorbed onto the beads. Compounds were re-volatilized by immersing the glass beads in hot water and were injected into a He carrier gas stream where the flow was split equally to 5 columns housed in 3 gas chromatographs (HP-6890).

The compounds were separated by gas chromatography and subsequently detected by electron capture (2 detectors), flame ionization (2), or quadrupole mass spectrometer (1) detectors. Peaks corresponding to compounds of interest were integrated manually. $CO/CO_2$ and $CH_4$ were analyzed using separate GC systems equipped with thermal conductivity and flame ionization detectors as described in Simpson et al., 2014. The $CO/CO_2$ GC-FID system is equipped with a Ni catalyst that converts CO into detectable $CH_4$.

2.3 Gas and $PM_{2.5}$ Emission factor calculations

Emission factors were calculated using the carbon-balance method, which assumes that all carbon in the fuel is converted to $CO_2$, CO, VOCs, and PM when the fuel is burned. The total gas-phase carbon emissions were approximated with the concentrations of $CO_2$, CO, as well as 76 detected gases measured using WAS. The mass of carbon in species $i$ ($m_{i,C}$) was calculated using equation (1).

$$m_{i,C} \, (g) = \frac{C_{i,C}(g/m^3)}{\sum C_{CO2,C} + C_{CO,C} + C_{CH4,C} + \cdots + C_{C6H6,C} \, (g/m^3)} \cdot m_{T,C} \, (kg) \cdot \frac{1000 \, g}{1 \, kg}$$    (1)

Where $C_{i,C}$ represents the mass concentration of carbon for species $i$, and $m_{T,C}$ refers to the carbon mass of the fuel, adjusted for ash and char carbon. The fraction of carbon in the fuel was taken to be 0.33 for buffalo dung and 0.45 for brushwood fuels based on Smith et al. 2000. Carbon in ash was estimated as 2.9% and 80.9% of the measured char mass for dry dung and dry brushwood, respectively (Smith et al., 2000b). Emissions factors for each species ($EF_i$) were calculated using equation (2).



$$EF_i\left(\frac{g\ VOC_i}{kg\ fuel}\right) = \frac{m_{i,C}(g) \cdot \dfrac{MW_i\ (g/mol)}{n_{c,i} \times 12.00\ (g/mol)}}{m_T(kg)} \tag{2}$$

Where $m_T$ is the net dry fuel burned for the cooking event, and $MW_i$ is the molecular weight of species $i$. The number of carbon atoms in molecule $i$ is denoted $n_{c,i}$. In addition to the emission factor normalized by the total fuel mass, emission factors were normalized to the total carbon mass in the fuel, calculated via equation (3).

$$EF_i\left(\frac{g\ VOC_i}{kg\ fuel\ C}\right) = EF_i\left(\frac{g\ VOC_i}{kg\ fuel}\right) \cdot \frac{m_T(kg)}{m_{T,C}(kg)} \tag{3}$$

$PM_{2.5}$ mass was determined gravimetrically using Teflon filters (PTFE, SKC Inc., 47 mm) weighed on a Cahn-28 electrobalance with a repeatability of ±1.0 µg after equilibrating for a minimum of 24 hours in a humidity and temperature-controlled environment both before and after sample collection (average temperature 19.5± 0.5 ºC; average relative humidity 49 ±5%). Another gravimetric filter was collected in the background during the cooking event and was equilibrated and weighed in the same way (Figure 1, Teflon C). Four field blanks filters were prepared by opening filters and then immediately closing and sealing the filters the same way as all samples; these filters had negligible mass loading (average: -0.75 µg) relative to samples (average: 1.57 mg). The background filter mass was adjusted to match the flow rate of the sample filter by assuming the flow rate is proportional to the filter mass. The background mass was then subtracted from the sample mass to obtain the mass of PM ($m_{PM}$) in equation (4) below.

$$\frac{EF_{PM}}{EF_{CO}} = \frac{m_{PM}/V_{air}}{m_{CO}/V_{air}} \tag{4}$$

2.4 Modified Combustion Efficiency (MCE)

Modified combustion efficiency (MCE) was defined as follows:

$$MCE = \frac{\Delta CO_2}{\Delta CO + \Delta CO_2} \tag{5}$$

where $\Delta CO$ and $\Delta CO_2$ are background-subtracted mixing ratios of CO and $CO_2$ for the time-integrated whole air sample. The total carbon mixing ratio is approximated by the sum of carbon monoxide and carbon dioxide in this definition.

2.5 SOA Forming potential

Relative SOA forming potential from measured VOCs was estimated using secondary organic aerosol potential (SOAP) values from Derwent et al., 2010, who used a photochemical transport model to simulate the SOA mass increase from the instantaneous emission of a particular VOC in a single parcel of air travelling across Europe. The model was outfitted with the Master Chemical Mechanism and the UK National Atmospheric Emission Inventory. SOA mass was estimated assuming equilibrium partitioning of oxidation products. Partitioning coefficients were calculated using absorptive partitioning theory of Pankow, 1994. All SOA mass increases from a particular VOC ($i$) were normalized to that of toluene and reported as SOAP values as shown in equation (6).



$$SOAP_i = \frac{\text{Increment in SOA mass concentration with species } i}{\text{Increment in SOA mass concentration with toluene}} \times 100 \tag{6}$$

SOA forming potential was calculated from the published SOAP values using equation (7).

$$\text{SOA potential} = \sum_{i=0}^{n} SOAP_i \times EF_i \left(\frac{\text{g VOC}_i}{\text{kg fuel C}}\right) \tag{7}$$

We emphasize that the SOA forming potential presented here is a relative value, and does not represent an absolute SOA yield.

2.6 OH reactivity

Total OH reactivity normalized by the mixing ratio of CO was calculated using equation (8).

$$\text{OH reactivity} \left(\frac{1}{\text{s·ppb CO}}\right) = \sum_{i=0}^{n} k_{OH,i} \left(\frac{\text{cm}^3}{\text{molec·s}}\right) \times ER_i \left(\frac{\text{ppt VOC}_i}{\text{ppb CO}}\right) \times 2.46 \times 10^7 \left(\frac{\text{molec}}{\text{cm}^3 \cdot \text{ppt}}\right) \tag{8}$$

Second-order rate constants ($k_{OH}$) at 25°C were taken from the NIST chemical kinetics database (Manion et al., 2015). $ER_i$ is the emission ratio for compound $i$ in ppt of VOC per ppb of CO. The last term serves as a conversion factor from VOC mixing ratio

to concentration at standard ambient temperature and pressure (25°C, 1 atm). By using the emission ratio to CO, we can track OH reactivity from VOCs depending on the extent of dilution from the plume. From here forward, the OH reactivity (s[-1]) reported is the average at the location of the sampling probes, or roughly 60 cm above the cookstove.

2.7 Ozone-forming potential (OFP)

The ozone-forming potential was estimated from the incremental reactivity of VOCs tabulated in Carter, 1994. Incremental

reactivities were calculated by comparing ozone formation before and after a VOC was introduced in a box model simulation. The Maximal Incremental Reactivity (MIR) scenario indicates that the chosen model inputs for $NO_x$ concentrations were optimized to yield the largest amount of $O_3$ production. In other words, $O_3$ production was not $NO_x$ limited. Because of this, the OFPs given here represent an upper limit for the $O_3$ production but reflect the rural villages where the measurements were performed. OFPs were calculated using equation (9).

$$\text{OFP} \left(\frac{\text{g O}_3}{\text{kg fuel C}}\right) = \sum_{i=0}^{n} MIR \left(\frac{\text{g O}_3}{\text{g VOC}_i}\right) \times EF_i \left(\frac{\text{g VOC}_i}{\text{kg fuel C}}\right) \tag{9}$$

2.8 Statistical analysis

One-way analysis of variance (ANOVA) with Tukey Post-Hoc testing was utilized to determine if there were significant differences ($p < 0.05$) in emissions of specific VOCs among categorical variables, i.e. stove and fuel types. All analyses were performed in R version 3.4.0 and RStudio version 1.0.143 (RStudio Inc, 2016).

**3 Results and discussion**

3.1 Chemical composition

Average VOC and $PM_{2.5}$ EFs (g/ kg dry fuel) as well as MCE are given in Table 1. The compounds are grouped by fuel-stove combination, with major species ($CO_2$, CO, $CH_4$, and $PM_{2.5}$) listed first, followed by sulfur-containing compounds, halogen-containing compounds, organonitrates, alkanes, alkenes, alkynes, aromatics, terpenes, and oxygenated compounds. The sample

size (n) used for calculating the average values and standard deviations was n=18 for dung-*chulha*, n=14 for brushwood-*chulha*,



n=13 for mixed-*chulha*, and n=10 for dung-*angithi*. For the majority of the compounds, the standard deviations are smaller than or comparable to the average values, indicating fair reproducibility.

Figure 2a visually shows the contribution of each compound class to the total gas-phase emissions on a grams per kilogram fuel carbon basis. The EFs are normalized by fuel carbon in Figure 2a in order to compare between cook fires generated with dung, wood, and wood-dung mixtures which have different carbon contents. The total measured VOC emissions from dung-*angithi* were roughly twice that of dung-*chulha* in terms of grams per kilogram fuel carbon. Further, dung-*chulha* emitted more than twice that of brushwood-*chulha*. The most prominent difference is non-furan oxygenates, making up almost half of all brushwood-*chulha* emissions and a smaller fraction for other fuel-stove combinations. While oxygenates make up a higher fraction of brushwood-*chulha* emissions, the absolute EFs for oxygenates from dung-burning and *angithi* cook fires are higher as discussed later in more detail.

Table 2 shows EFs (g/ kg fuel C) for select VOCs. The differences in mean EFs for each fuel-stove combination are also included in Table 2. Mean differences in EFs reported for *chulha* and *angithi* stoves were calculated for cook fires utilizing only dung fuels. Likewise, mean EFs for wood and dung cook fires only represent cooking events using the *chulha*. This was done to isolate a single variable—either fuel or stove type. For all alkanes and most alkenes, we measured higher emissions for dung-*angithi* cook fires (Table 2). Also, from the mean differences in EFs, we found that stove specific combustion conditions impacts emissions more than the selection of fuel type. The difference is so dramatic for alkanes and most alkenes that the mean difference in EFs for cookstoves burning dung is always larger than the mean EF of that compound. For comparison, the mean difference in EFs for *chulha* cookstoves is always lower than the overall mean EF. Ethene was an exception; there was no relationship between ethene emissions and stove type. On the other hand, the mean EF of ethene by dung cook fires was very large compared to mean EFs from brushwood cook fires with a mean difference in EFs of 4.05 g/ kg fuel C. Some alkenes with two double bonds were also exceptions. For 1,3-butadiene (p= 0.06) and 1,2-butadiene (p= 0.089), stove and EF may or may not have a significant relationship. 1,2-Propadiene emissions from *chulha* cookstoves are higher (p< 0.01). All three compounds still show a significant relationship to fuel type with EFs being higher for dung cook fires.

Similar to alkanes and alkenes, aromatics, oxygenates, halogen-, and sulfur-containing compounds all had higher emissions per kilogram of fuel carbon when dung fuels and *angithi* stoves were utilized compared to brushwood fuels and *chulha* stoves, respectively. Plastic bags were often used to start the cook fire, which could be a source of chlorine-containing compounds. Here and in the next paragraph, we focus on exceptional compounds. Interestingly, benzene had higher emissions from *chulha* stoves, which had higher MCEs when cooking with dung fuels compared to *angithi* stoves (dung-*chulha* 3.18 g/ kg fuel C and dung-*angithi* 2.38 g/ kg fuel C). As the simplest aromatic compound, benzene also had the largest average difference in fuel type EFs compared to other aromatics (2.18 g/ kg fuel C, dung-wood). This information is relevant for exposure assessment, as benzene is a known human carcinogen. While the cook usually cannot control the stove used, as the *angithi* and *chulha* are used to prepare different types of meals, and exposure to benzene is not straightforward from its emission factors, it is a notable result of potential concern in regards to public health.

Higher emissions of alkynes were observed from dung fuels and *chulha* cookstoves. The latter observation is consistent with the literature showing flaming combustion generates more alkynes (Barrefors and Petersson, 1995; Lee et al., 2005). *Chulha* cook fires always had higher MCE than *angithi* cook fires (Table 1) which rely on smoldering combustion. Approximately the same difference in alkyne emissions results from comparing the *chulha* to the *angithi* using dung, in relation to using wood versus dung in combination with the *chulha*. There were two exceptions in stove type for 1-butane (p= 0.055) and 2-butane (p>> 0.05).





The former may or may not have a relationship with stove type, while the latter does not. Emissions of some compounds did not show a relationship with either fuel or stove type, and are listed in Table S1.1.

VOC emissions from Stockwell et al., 2016 are also provided in Table 1 for comparison of VOC EFs. Samples in Stockwell et al., 2016 were collected in April 2015 in and around Kathmandu and the Tarai plains, which border India. While both are EFs

from cookstoves using similar fuels, there are differences in the studies that should be noted. Stockwell et al., 2016 collected measurements of simulated cooking in a laboratory and from cooking fires in households; it was not noted in the latter case what meals were cooked. EFs were calculated from WAS measurements, but as grab samples in an area of the kitchen away from the fire (as opposed to the time-integrated approach used here). Emissions were assumed to be well-mixed in the kitchen prior to sampling. Stockwell et al., 2016 also used a range of stoves, including the traditional single-pot mud stove, open three-stone fire,

*bhuse chulo*, rocket, chimney, and forced draft stoves. "Dung" cook fires sometimes used a combination of fuels, such as wood. Finally, our study also has a larger sample size than Stockwell et al., 2016 with n=49 versus n≈10.

The emission factors for most compounds determined in this study were lower compared to those reported by Stockwell et al., 2016. Figure S2.1 visually shows that the EFs were generally lower in the present study. In some cases, EFs in this study were an order of magnitude lower, most notably n-pentane and n-hexane. We also found that our emission factors were always higher for

dung-*chulha* compared to brushwood-*chulha,* which was not always the case in Stockwell et al., 2016. The EFs in Stockwell et al., 2016 could be biased high due to calculations rather than real differences in emissions. For example, ignoring ash and char carbon and using the same carbon content inflates the EFs reported in our paper by 7% for dung and 24% for brushwood emissions. However, this is a small percentage compared to the observed differences between EFs between the two measurements. Therefore, it is likely that there were real differences in emissions due to simulated cooking, different cooking

activities, and/or the stoves utilized. Roden et al., 2009 and Johnson et al., 2008 showed that cooking activities can strongly influence emissions, for example due to the cook tending to the cook fire differently, thus affecting combustion conditions.

3.2 Modified combustion efficiency

The use of dung and *angithi*, rather than brushwood and *chulha*, respectively, results in lower modified combustion efficiencies as shown in Figure 3. In general, at lower MCEs we measured higher emissions of gas-phase compounds as discussed in Section

3.1. For example, emissions of ethane (Figure 3a) and other alkanes increase with decreasing MCE, however not in a linear manner; in other studies a linear regression analysis is used to convey a robust correlation (Liu et al., 2017; Selimovic et al., 2018). For other VOCs, the dependence of the EF on MCE is more complicated, with the maximum EF observed at intermediate MCE values. For example, the ethene EF (Figure 3b) increases with decreasing MCE at MCEs > 0.85, but it has the opposite trend at MCEs < 0.85. Previously, we discussed that there is no relationship between ethene EF and stove type and we see this

more clearly in Figure 3b. Alkynes have the same relationship to MCE as ethene, but it is even more pronounced (Figure 3c). Benzene (Figure 3e) stands apart from other aromatics with a relationship with MCE similar to ethene's, while other aromatics have an EF versus MCE curve similar to alkanes and most other VOCs. In Figure 3e, we see again that emissions from brushwood-*chulha* and dung-*angithi* cook fires result in lower emissions of benzene compared to dung-*chulha*. Alkenes with two double bonds generally have a negative correlation between emissions and MCE, such as 1,3-butadiene in Figure 3f. The 1,3-

butadiene EF versus MCE plot is not necessarily representative of all analogous plots for alkenes with two double bonds, as they have different shapes. 1,3-butadiene was chosen as its emission is high compared to other compounds in its subcategory and it also has health implications. It also happens to have a more linear relationship with MCE, albeit noisy.





It is of interest to compare EFs obtained from different fuel-stove combinations but with the same MCE. In the case of ethane, different cook fire types yield vastly different EFs at the same MCE. For example, at MCE≈ 0.87, mixed-*chulha* has an EF of roughly 1.5 g/ kg fuel C, dung-*chulha* is 2.5 g/ kg fuel C, and dung-*angithi* is 5.5 g/ kg fuel C. Knowledge of the cook fire MCE alone is not sufficient to determine the EF of ethane. A similar conclusion can be reached for most of the measured gases,

including non-ethene alkenes in Figure 3d.

3.3 Secondary pollutant formation and reactivity

*OH reactivity and ozone forming potential*

Total OH reactivity ($s^{-1}$) based on the measured VOCs in Figure 2b is given per ppb of CO. Predicted OH reactivities in the village due to a single cooking event are 10.2, 6.73, 4.93, and 4.83 $s^{-1}$ for emissions from dung-*angithi*, dung-*chulha*, mixed-

*chulha*, and brushwood-*chulha* cook fires, respectively. This assumes a CO mixing ratio of 338 ppb, which we measured as the average background mixing ratio over the whole campaign. The relative total OH reactivity is over twice as high for dung-*angithi* cook fires as it is for brushwood-*chulha* cook fires.

The classes of compounds that act as the most important OH radical sinks in descending order are alkenes, oxygenates, furans, terpenes, and aromatics. Alkenes make up more than 50% of OH reactivity for all cook fire types. Ethene (by fuel type) and

propene (by fuel and stove combination) are mostly responsible for the differences in fuel-stove combination results for alkenes. For oxygenates, methanol ($p< 0.01$) and acrolein ($p< 0.05$) have significantly higher OH reactivity with wood fuel, while acetaldehyde has significantly higher OH reactivity with the *angithi* stove ($p< 0.001$). Differences in OH reactivity due to furans were observed for stove type but not fuel type. All three of the measured furans significantly contribute ($p<0.001$) to a 6% increase in the fraction of OH reactivity due to furans for dung-*angithi* (12%) as opposed to dung-*chulha* (6%). The percentage

of OH reactivity due to aromatics is constant at ~4% for the fuel-stove combinations. However, different aromatic compounds are responsible for this ~4% contribution depending on the cook fire type. Benzene dominates OH reactivity due to aromatics for *chulha* cook fires. For *angithi* cook fires aromatics other than benzene, in particular toluene, dictate the OH reactivity for aromatics. Isoprene is solely responsible for the differences in OH reactivity due to terpenes.

Figure 2d shows the total ozone forming potential (g/ $O_3$ kg fuel) in the MIR scenario, as well as contributions to OFP by

compound class. A critical step in photochemical ozone production is VOC reacting with OH. Therefore, the ozone forming potential contributions by compound class are similar to those for OH reactivity (Figure 2c). Total OFP is nearly a factor of 3 higher for dung-*chulha* compared to brushwood-*chulha,* while it's twice as high for dung-*angithi* as compared to dung-*chulha*.

*SOA formation potential*

SOAP-weighted emissions relate SOA production from the different cook fire types in a qualitative manner. The contribution of

each compound class to the total SOA formation potential is shown in Figure 2c. The sum of the contributions by each compound class, or the total SOA forming potential, is also shown below the pie charts. Dung fuels and *angithi* stoves yield larger amounts of SOA. However, fuel type is more important than stove type in terms of SOA formation. SOAP-weighted emissions are a factor of three higher for dung-*chulha* compared to brushwood-*chulha*. We discussed previously that benzene emissions are significantly higher from *chulha* cook fires compared to *angithi* cook fires. These higher benzene emissions

directly impact public health and also dictate SOA formation for *chulha* emissions. Benzene emissions are responsible for at least half of the SOA formation from the *chulha* cook fire VOCs we measured. Beyond benzene, aromatics make up on average




roughly 95% of SOA precursors for all cook fires. While benzene is prominent for *chulha* cook fires, $C_8$-$C_9$ aromatics, toluene, and benzene contribute in approximately equal proportions to SOA formation in dung-*angithi* smoke plumes.

**4 Atmospheric implications and conclusions**

5 The extent of ozone formation hinges on the villages' overall $NO_x$ levels. However, in a VOC-limited regime, with each household in this village cooking three meals a day using the *chulha* and mixed fuels (brushwood + dung), $3.3 \times 10^5$ g ozone per day is expected to be produced. This was estimated based off the Census of India 2011 data for the village of Khatela and assuming the same fuel consumption as that used in this study. Over the lunch hour, when solar radiation is most intense, 30 ppb ozone is predicted. For this calculation we assumed a calm wind speed of 0.5 m/s, and confined our analysis to the village of Khatela with a boundary layer height of 1 km and village length of 1 km. In a similar way we calculated the amount of ozone 10 that could be generated from cooking animal fodder. We assumed that every household prepares animal fodder every three days in addition to the assumptions already discussed, resulting in an additional $7.9 \times 10^4$ grams of $O_3$ produced per day. If we assume every household in the village prepares animal fodder in the same hour, ozone levels of 7 ppb are predicted, using the same assumptions described earlier for the lunch hour. We should note that these estimations are approximate and a regional air quality model with detailed household level inputs should be used to more precisely predict the impact of cook fire emissions on 15 ozone levels.

Using dung patties as opposed to brushwood has a large impact on local $PM_{2.5}$ and ozone levels. Measured $PM_{2.5}$ concentrations were more than a factor of two higher for dung-*chulha* compared to brushwood-*chulha* in grams emitted per kilogram of fuel carbon burned. In addition to this, the total SOA forming potential is three times higher for dung-*chulha* than that of brushwood-*chulha*. We also estimated that dung-*chulha* cook fires produce roughly 3 times more ozone in the MIR regime than brushwood-20 *chulha* cook fires (163 g $O_3$ kg fuel C versus 56.9 g $O_3$ kg fuel C). However, compounds such as benzene are emitted in higher quantities from the *chulha* (1.03 g $kg^{-1}$ dry fuel) versus *angithi* (0.373 g $kg^{-1}$ dry fuel), and this public health concern should be investigated in more detail.

**Acknowledgements**

25 We thank the village of Khatela and our cook for welcoming us and for participating in the study. We also want to acknowledge Sneha Gautam's role in supporting the field work. This research was supported by EPA STAR grant R835425 Impacts of household sources on outdoor pollution at village and regional scales in India. The contents are solely the responsibility of the authors and do not necessarily represent the official views of the US EPA. The US EPA does not endorse the purchase of any commercial products or services mentioned in the publication.

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





**Figure 1. Sampling train for collecting cookstove emissions. PCXR8 (blue) are sampling pumps, WAS or whole air samples (green) are the air samplers, and orange boxes are Teflon or quartz filters used to collect PM$_{2.5}$.**

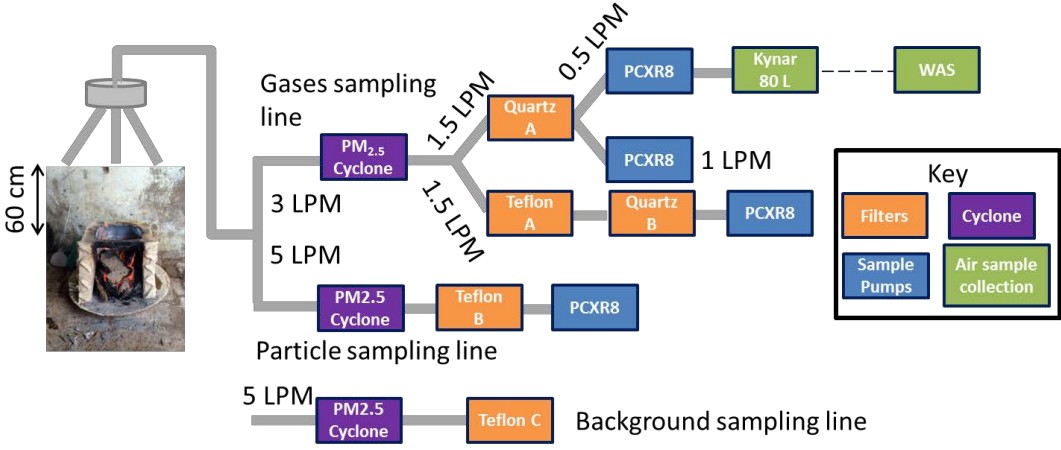





**Figure 2. Pie charts showing the contribution of each species class to gas-phase composition (a), OH reactivity (b), SOAP-weighted emissions (c), and ozone-forming potential (d). For (b) and (d), total aromatics are shown rather than the breakdown of aromatics shown in (a) and (c). Sums of all components are shown below the pie chart.**

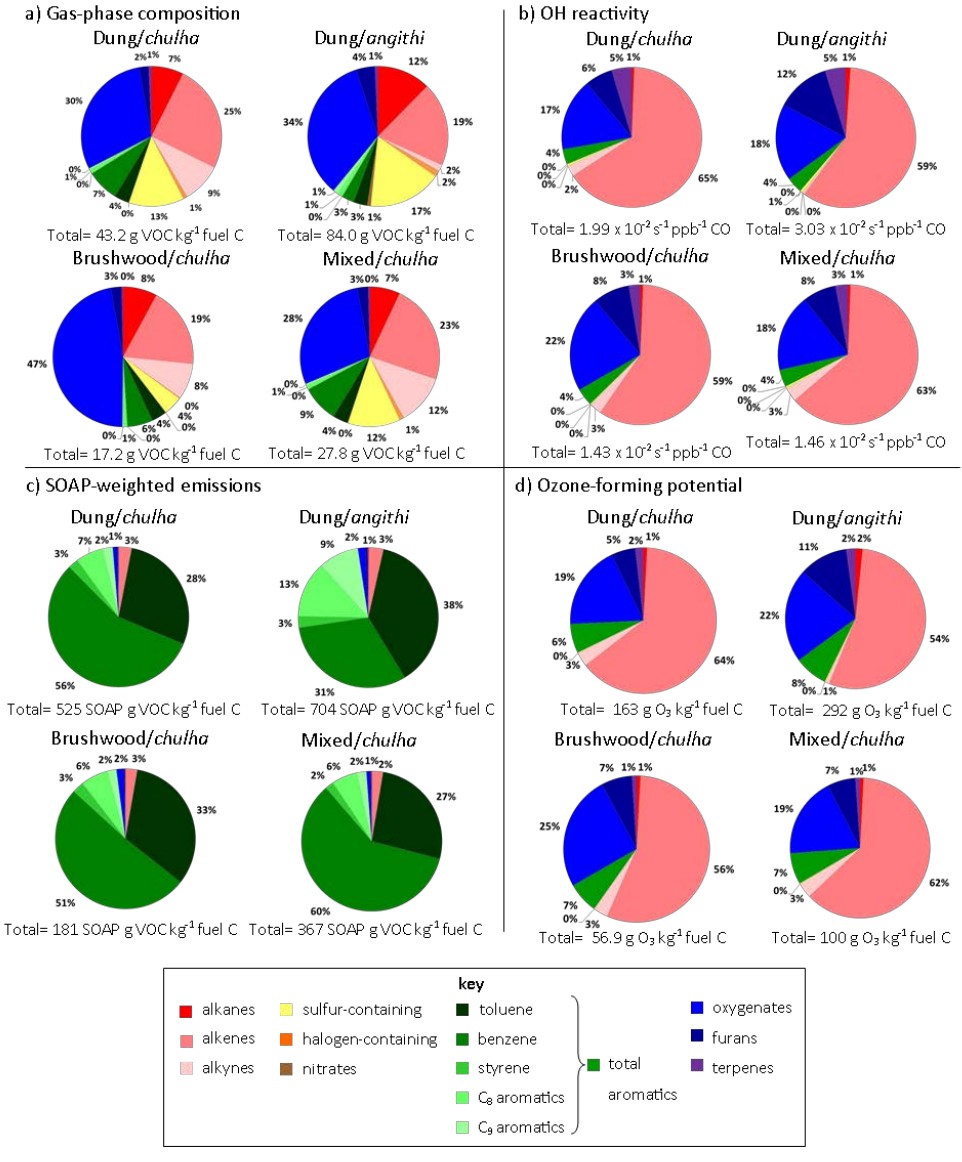

[a]1-Buten-3-yne is grouped in with alkynes



**Figure 3 (left). Emission factors as a function of MCE for select species. Open circles indicate cooking events conducted with *angithi* stoves, whereas filled squares indicate *chulha* stoves. Color indicates fuel, either brushwood (blue), dung (red), or mixed (purple).**

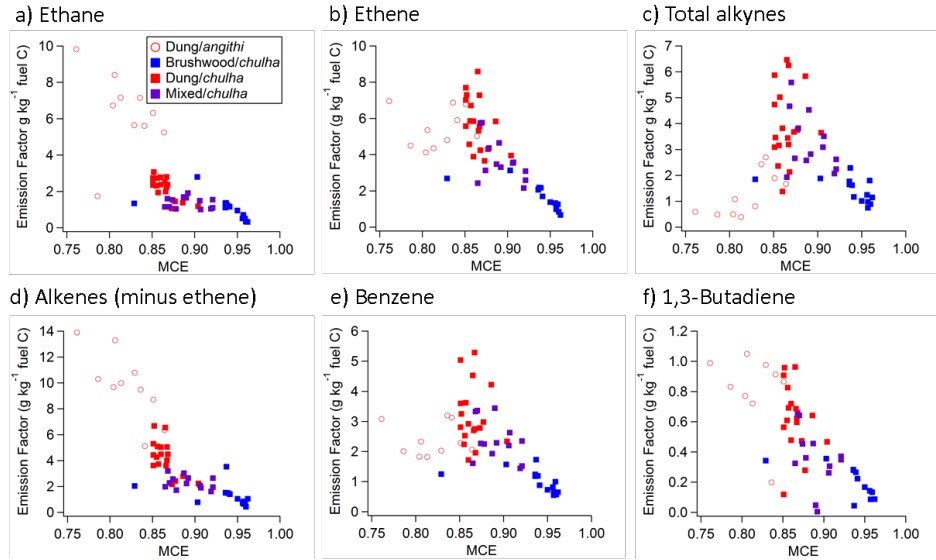

[a]1-Buten-3-yne is grouped in with alkynes





**Table 1.** Averaged emission factors and standard deviation of $PM_{2.5}$ and gas-phase species (g kg$^{-1}$ dry fuel) for dung-*chulha*, brushwood-*chulha*, mixed-*chulha*, and dung-*angithi* cook fires. Previously published emission factors (g kg$^{-1}$ dry fuel) from dung and hardwood cook fires are shown for comparison (Stockwell et al. 2016). Sample sizes for the current study (n) were n=18 for dung-*chulha*, n=14 for brushwood-*chulha*, n=13 for mixed-*chulha*, and n=10 for dung-*angithi*.

| Compound (formula) | Dung-*chulha* Average (SD) | Brushwood-*chulha* Average (SD) | Mixed-*chulha* Average (SD) | Dung-*angithi* Average (SD) | Stockwell et al. (2016) Dung Average (SD) | Stockwell et al. (2016) Hardwood Average (SD) |
|---|---|---|---|---|---|---|
| MCE | 0.865 (0.014) | 0.937 (0.035) | 0.892 (0.021) | 0.819 (0.031) | 0.898 | 0.923 |
| $PM_{2.5}$ | 19.2 (7.1) | 7.42 (5.67) | 11.0 (2.0) | 33.2 (7.6) | 14.73 (0.33)* | 7.97 (3.80)* |
| Carbon dioxide ($CO_2$) | 984 (23) | 1242 (61) | 969 (31) | 888 (48) | 1129 (80) | 1462 (16) |
| Carbon monoxide (CO) | 97.7 (9.5) | 53.0 (30.1) | 74.8 (16.0) | 125 (20) | 80.9 (13.8) | 77.2 (13.5) |
| Methane ($CH_4$) | 6.92 (1.23) | 4.80 (2.09) | 4.84 (0.89) | 15.1 (2.6) | 6.65 (0.46) | 5.16 (1.39) |
| **Sulfur-containing** | | | | | | |
| Carbonyl sulfide (OCS) | 0.124 (0.040) | $1.44 (0.54) \times 10^{-2}$ | $8.50 (2.42) \times 10^{-2}$ | 0.352 (0.217) | 0.148 (0.123) | $1.87 (1.15) \times 10^{-2}$ |
| DMS ($C_2H_6S$) | $9.69 (4.54) \times 10^{-3}$ | $1.39 (1.34) \times 10^{-3}$ | $4.81 (2.26) \times 10^{-3}$ | $4.34 (3.11) \times 10^{-2}$ | $2.37 (0.08) \times 10^{-2}$ | 0.255 (0.359) |
| **Halogen-containing** | | | | | | |
| Dichloromethane ($CH_2Cl_2$) | $4.46 (3.94) \times 10^{-4}$ | $2.18 (3.13) \times 10^{-4}$ | $4.04 (6.44) \times 10^{-4}$ | $4.56 (2.73) \times 10^{-4}$ | nm | nm |
| Chloromethane ($CH_3Cl$) | 1.78 (0.70) | 0.280 (0.157) | 1.02 (0.42) | 4.58 (1.89) | 1.60 (1.53) | $2.36 (1.62) \times 10^{-2}$ |
| Bromomethane ($CH_3Br$) | $6.57 (2.78) \times 10^{-3}$ | $7.92 (2.13) \times 10^{-4}$ | $4.35 (1.81) \times 10^{-3}$ | $1.43 (0.57) \times 10^{-2}$ | $5.34 (3.02) \times 10^{-3}$ | $5.61 (3.01) \times 10^{-4}$ |
| Iodomethane ($CH_3I$) | $6.10 (4.78) \times 10^{-4}$ | $9.62 (2.31) \times 10^{-5}$ | $2.41 (0.66) \times 10^{-4}$ | $8.83 (1.62) \times 10^{-4}$ | $4.39 (1.78) \times 10^{-4}$ | $1.23 (1.11) \times 10^{-4}$ |
| Ethyl chloride ($C_2H_5Cl$) | $2.54 (1.17) \times 10^{-3}$ | $4.22 (3.72) \times 10^{-4}$ | $1.59 (0.67) \times 10^{-3}$ | $9.11 (3.50) \times 10^{-3}$ | nm | nm |
| Dichloroethane ($C_2H_4Cl_2$) | $8.80 (2.98) \times 10^{-4}$ | $2.55 (2.17) \times 10^{-4}$ | $1.21 (2.32) \times 10^{-3}$ | $1.47 (0.91) \times 10^{-3}$ | $4.97 \times 10^{-3}$ (-) | $1.24 (0.30) \times 10^{-4}$ |
| **Nitrates** | | | | | | |
| Methyl nitrate ($CH_3ONO_2$) | $1.83 (5.18) \times 10^{-3}$ | $5.34 (14.4) \times 10^{-3}$ | $6.60 (11.7) \times 10^{-3}$ | 0.170 (0.339) | $1.46 (1.94) \times 10^{-2}$ | $6.96 (5.73) \times 10^{-3}$ |
| Ethyl nitrate ($CH_3ONO_2$) | $2.37 (3.86) \times 10^{-4}$ | $5.54 (10.2) \times 10^{-4}$ | $2.27 (6.40) \times 10^{-3}$ | $4.53 (11.6) \times 10^{-2}$ | nm | nm |
| i-Propylnitrate ($C_3H_7ONO_2$) | $1.90 (1.61) \times 10^{-4}$ | $2.40 (4.92) \times 10^{-4}$ | $4.10 (8.38) \times 10^{-4}$ | $5.90 (12.1) \times 10^{-3}$ | nm | nm |
| n-Propylnitrate ($C_3H_7ONO_2$) | $6.32 (5.23) \times 10^{-5}$ | $9.01 (14.1) \times 10^{-5}$ | $1.44 (3.25) \times 10^{-4}$ | $1.82 (4.35) \times 10^{-3}$ | nm | nm |





| | | | | | | |
|---|---|---|---|---|---|---|
| 2-Butylnitrate $(C_4H_9ONO_2)$ | $2.69\ (2.14)\times 10^{-4}$ | $1.05\ (1.13)\times 10^{-4}$ | $7.10\ (20.3)\times 10^{-4}$ | $2.45\ (4.09)\times 10^{-3}$ | nm | nm |
| 3-Pentylnitrate $(C_5H_{11}ONO_2)$ | $4.75\ (1.61)\times 10^{-5}$ | $2.29\ (2.08)\times 10^{-5}$ | $3.13\ (1.97)\times 10^{-5}$ | $1.94\ (4.06)\times 10^{-4}$ | nm | nm |
| 2-Pentylnitrate $(C_5H_{11}ONO_2)$ | $2.37\ (2.10)\times 10^{-5}$ | $1.63\ (2.46)\times 10^{-5}$ | $1.25\ (1.26)\times 10^{-5}$ | $1.82\ (4.54)\times 10^{-4}$ | nm | nm |
| **Alkanes** | | | | | | |
| Ethane $(C_2H_6)$ | 0.717 (0.193) | 0.380 (0.247) | 0.422 (0.096) | 2.06 (0.69) | 1.08 (0.30) | 0.160 (0.122) |
| Propane $(C_3H_8)$ | 0.211 (0.073) | $9.48\ (8.41)\times 10^{-2}$ | 0.116 (0.032) | 0.819 (0.157) | 0.457 (0.137) | 0.202 (0.140) |
| i-Butane $(C_4H_{10})$ | $1.73\ (0.71)\times 10^{-2}$ | $4.60\ (4.86)\times 10^{-3}$ | $9.51\ (2.75)\times 10^{-3}$ | $7.27\ (1.54)\times 10^{-2}$ | 0.215 (0.126) | 0.406 (0.478) |
| n-Butane $(C_4H_{10})$ | $4.71\ (1.88)\times 10^{-2}$ | $1.57\ (1.67)\times 10^{-2}$ | $2.68\ (0.88)\times 10^{-2}$ | 0.215 (0.047) | 0.29 (0.09) | 1.11 (1.48) |
| n-Pentane $(C_5H_{12})$ | $2.01\ (0.98)\times 10^{-2}$ | $4.44\ (4.08)\times 10^{-3}$ | $9.12\ (3.71)\times 10^{-3}$ | $6.80\ (2.95)\times 10^{-2}$ | 0.190 (0.254) | $2.18\ (1.73)\times 10^{-2}$ |
| n-Hexane $(C_6H_{14})$ | $1.03\ (0.47)\times 10^{-2}$ | $1.96\ (1.58)\times 10^{-3}$ | $5.31\ (1.87)\times 10^{-3}$ | $4.93\ (1.10)\times 10^{-2}$ | 0.291 (0.248) | $1.85\times 10^{-2}\ (-)$ |
| n-Heptane $(C_7H_{16})$ | $7.21\ (3.43)\times 10^{-3}$ | $9.23\ (6.94)\times 10^{-4}$ | $3.92\ (1.23)\times 10^{-3}$ | $3.17\ (0.85)\times 10^{-2}$ | 0.114 (0.069) | $1.01\ (1.35)\times 10^{-2}$ |
| 2-Methylpentane $(C_6H_{14})$ | $6.21\ (2.81)\times 10^{-3}$ | $1.23\ (0.99)\times 10^{-3}$ | $2.57\ (1.61)\times 10^{-3}$ | $2.29\ (1.67)\times 10^{-2}$ | 0.231 (0.192) | $9.93\ (12.9)\times 10^{-3}$ |
| 3-Methylpentane $(C_6H_{14})$ | $3.71\ (1.70)\times 10^{-3}$ | $1.21\ (1.01)\times 10^{-3}$ | $1.57\ (0.76)\times 10^{-3}$ | $7.54\ (4.30)\times 10^{-3}$ | 0.155 (0.137) | $6.79\ (6.63)\times 10^{-3}$ |
| **Alkenes** | | | | | | |
| Ethene $(C_2H_4)$ | 1.86 (0.48) | 0.626 (0.284) | 1.13 (0.38) | 1.77 (0.35) | 4.23 (1.39) | 2.70 (1.17) |
| Propene $(C_3H_6)$ | 0.807 (0.235) | 0.286 (0.202) | 0.417 (0.091) | 1.61 (0.33) | 1.47 (0.58) | 0.576 (0.195) |
| 1-Butene $(C_4H_8)$ | 0.158 (0.047) | $6.32\ (4.59)\times 10^{-2}$ | $8.38\ (1.83)\times 10^{-2}$ | 0.366 (0.096) | 0.399 (0.331) | 0.726 (0.904) |
| i-Butene $(C_4H_8)$ | 0.133 (0.057) | $3.46\ (2.50)\times 10^{-2}$ | $6.40\ (1.86)\times 10^{-2}$ | 0.353 (0.158) | 0.281 (0.091) | 0.846 (1.113) |
| trans-2-Butene $(C_4H_8)$ | $4.45\ (1.60)\times 10^{-2}$ | $2.00\ (1.27)\times 10^{-2}$ | $2.38\ (0.70)\times 10^{-2}$ | 0.151 (0.055) | 0.151 (0.010) | $6.78\ (5.98)\times 10^{-2}$ |
| cis-2-Butene $(C_4H_8)$ | $3.38\ (1.19)\times 10^{-2}$ | $1.51\ (0.95)\times 10^{-2}$ | $1.80\ (0.52)\times 10^{-2}$ | 0.107 (0.047) | 0.102 (0.016) | $5.51\ (4.76)\times 10^{-2}$ |
| 3-Methyl-1-butene $(C_5H_{10})$ | $1.46\ (0.48)\times 10^{-2}$ | $5.74\ (4.49)\times 10^{-3}$ | $7.30\ (1.94)\times 10^{-3}$ | $3.82\ (0.88)\times 10^{-2}$ | $5.58\ (3.50)\times 10^{-2}$ | $7.43\ (5.79)\times 10^{-3}$ |
| 2-Methyl-1-butene $(C_5H_{10})$ | $2.71\ (1.28)\times 10^{-2}$ | $9.96\ (10.9)\times 10^{-3}$ | $1.19\ (0.42)\times 10^{-2}$ | $7.70\ (3.99)\times 10^{-2}$ | nm | nm |
| 2-Methyl-2-butene $(C_5H_{10})$ | $2.51\ (1.26)\times 10^{-2}$ | $6.40\ (4.78)\times 10^{-3}$ | $1.10\ (0.47)\times 10^{-2}$ | $9.17\ (4.70)\times 10^{-2}$ | nm | nm |
| 1-Pentene $(C_5H_{10})$ | $4.17\ (1.59)\times 10^{-2}$ | $9.65\ (6.55)\times 10^{-3}$ | $2.13\ (0.60)\times 10^{-2}$ | 0.122 (0.033) | 0.168 (0.086) | $1.43\ (0.94)\times 10^{-2}$ |
| trans-2-Pentene $(C_5H_{10})$ | $1.74\ (0.65)\times 10^{-2}$ | $8.89\ (5.77)\times 10^{-3}$ | $8.69\ (2.22)\times 10^{-3}$ | $5.14\ (2.70)\times 10^{-2}$ | 0.115 (0.035) | $1.05\ (0.83)\times 10^{-2}$ |



| | | | | | | |
|---|---|---|---|---|---|---|
| cis-2-Pentene ($C_5H_{10}$) | $1.00\ (0.36) \times 10^{-2}$ | $5.55\ (3.62) \times 10^{-3}$ | $4.98\ (1.26) \times 10^{-3}$ | $2.50\ (1.28) \times 10^{-2}$ | $5.14\ (0.76) \times 10^{-2}$ | $8.69 \times 10^{-3}$ (-) |
| 1-Hexene ($C_6H_{12}$) | $6.10\ (2.46) \times 10^{-2}$ | $1.26\ (0.73) \times 10^{-2}$ | $3.09\ (0.91) \times 10^{-2}$ | $0.167\ (0.050)$ | nm | nm |
| 1,2-Propadiene ($C_3H_4$) | $3.76\ (1.69) \times 10^{-2}$ | $1.31\ (0.62) \times 10^{-2}$ | $2.32\ (0.86) \times 10^{-2}$ | $1.80\ (0.923) \times 10^{-2}$ | $7.15\ (6.76) \times 10^{-2}$ | $2.33\ (1.07) \times 10^{-2}$ |
| 1,2-Butadiene ($C_4H_6$) | $5.54\ (1.68) \times 10^{-3}$ | $2.82\ (1.81) \times 10^{-3}$ | $3.10\ (1.06) \times 10^{-3}$ | $4.33\ (1.59) \times 10^{-3}$ | nm | nm |
| 1,3-Butadiene ($C_4H_6$) | $0.203\ (0.071)$ | $7.44\ (3.99) \times 10^{-2}$ | $0.108\ (0.061)$ | $0.263\ (0.082)$ | $0.409\ (0.306)$ | $0.204\ (0.144)$ |
| Isoprene ($C_5H_8$) | $8.94\ (5.80) \times 10^{-2}$ | $1.98\ (1.48) \times 10^{-2}$ | $3.03\ (2.39) \times 10^{-2}$ | $0.188\ (0.143)$ | $0.325\ (0.443)$ | $4.16\ (2.23) \times 10^{-2}$ |
| 1,3-Pentadiene ($C_5H_8$) | $1.96\ (1.05) \times 10^{-2}$ | $9.17\ (4.79) \times 10^{-3}$ | $9.39\ (6.43) \times 10^{-3}$ | $5.66\ (2.94) \times 10^{-2}$ | nm | nm |
| **Alkynes** | | | | | | |
| Ethyne | $1.13\ (0.42)$ | $0.467\ (0.160)$ | $0.890\ (0.323)$ | $0.325\ (0.238)$ | $0.593\ (0.443)$ | $0.764\ (0.363)$ |
| 1-Propyne | $9.42\ (3.46) \times 10^{-2}$ | $3.82\ (1.76) \times 10^{-2}$ | $5.99\ (2.22) \times 10^{-2}$ | $5.20\ (2.83) \times 10^{-2}$ | nm | nm |
| 1-Buten-3-yne ($C_4H_4$) | $5.04\ (1.72) \times 10^{-2}$ | $1.86\ (0.90) \times 10^{-2}$ | $3.46\ (1.53) \times 10^{-2}$ | $1.74\ (1.26) \times 10^{-2}$ | nm | nm |
| 1-Butyne ($C_4H_6$) | $7.72\ (2.29) \times 10^{-3}$ | $4.07\ (2.24) \times 10^{-3}$ | $4.48\ (1.41) \times 10^{-3}$ | $5.97\ (1.93) \times 10^{-3}$ | $2.29\ (1.38) \times 10^{-2}$ | $1.28\ (0.47) \times 10^{-2}$ |
| 2-Butyne ($C_4H_6$) | $4.31\ (1.15) \times 10^{-3}$ | $2.55\ (1.44) \times 10^{-3}$ | $2.47\ (0.70) \times 10^{-3}$ | $4.52\ (1.40) \times 10^{-3}$ | $1.86\ (0.91) \times 10^{-2}$ | $1.02\ (0.66) \times 10^{-2}$ |
| 1,3-Butadyne ($C_4H_2$) | $6.07\ (2.66) \times 10^{-3}$ | $2.71\ (1.21) \times 10^{-3}$ | $5.43\ (2.01) \times 10^{-3}$ | $1.53\ (1.31) \times 10^{-3}$ | nm | nm |
| **Aromatics** | | | | | | |
| Benzene ($C_6H_6$) | $1.03\ (0.33)$ | $0.373\ (0.149)$ | $0.723\ (0.218)$ | $0.769\ (0.175)$ | $1.96\ (0.45)$ | $1.05\ (0.19)$ |
| Toluene ($C_7H_8$) | $0.483\ (0.273)$ | $0.221\ (0.085)$ | $0.297\ (0.077)$ | $0.860\ (0.167)$ | $1.26\ (0.05)$ | $0.241\ (0.160)$ |
| Ethylbenzene ($C_8H_{10}$) | $3.41\ (0.791) \times 10^{-2}$ | $1.25\ (1.20) \times 10^{-2}$ | $1.97\ (0.40) \times 10^{-2}$ | $9.78\ (1.66) \times 10^{-2}$ | $0.366\ (0.085)$ | $4.19\ (4.25) \times 10^{-2}$ |
| m/p-Xylene ($C_8H_{10}$) | $6.36\ (1.26) \times 10^{-2}$ | $2.78\ (1.56) \times 10^{-2}$ | $4.03\ (0.98) \times 10^{-2}$ | $0.148\ (0.030)$ | $0.601\ (0.294)$ | $9.57\ (7.99) \times 10^{-2}$ |
| o-Xylene ($C_8H_{10}$) | $2.38\ (0.76) \times 10^{-2}$ | $8.37\ (5.78) \times 10^{-3}$ | $1.44\ (0.41) \times 10^{-2}$ | $7.96\ (1.91) \times 10^{-2}$ | $0.228\ (0.083)$ | $3.93\ (4.31) \times 10^{-2}$ |
| Styrene ($C_8H_8$) | $5.88\ (1.58) \times 10^{-2}$ | $2.28\ (1.50) \times 10^{-2}$ | $3.40\ (1.90) \times 10^{-2}$ | $8.63\ (5.96) \times 10^{-2}$ | $0.255\ (0.091)$ | $8.71\ (6.69) \times 10^{-2}$ |
| i-Propylbenzene ($C_9H_{12}$) | $2.91\ (0.77) \times 10^{-3}$ | $1.20\ (1.11) \times 10^{-3}$ | $1.69\ (0.45) \times 10^{-3}$ | $9.30\ (4.90) \times 10^{-3}$ | $1.87\ (1.40) \times 10^{-2}$ | $1.70\ (1.67) \times 10^{-2}$ |
| n-Propylbenzene ($C_9H_{12}$) | $6.48\ (2.59) \times 10^{-3}$ | $1.84\ (1.65) \times 10^{-3}$ | $4.02\ (1.59) \times 10^{-3}$ | $3.95\ (2.69) \times 10^{-2}$ | $3.10\ (1.45) \times 10^{-2}$ | $1.78\ (1.58) \times 10^{-2}$ |
| 3-Ethyltoluene ($C_9H_{12}$) | $1.44\ (0.48) \times 10^{-2}$ | $5.46\ (4.40) \times 10^{-3}$ | $8.59\ (3.26) \times 10^{-3}$ | $7.14\ (4.13) \times 10^{-2}$ | $5.61\ (2.38) \times 10^{-2}$ | $2.62\ (0.54) \times 10^{-2}$ |
| 4-Ethyltoluene ($C_9H_{12}$) | $6.35\ (2.36) \times 10^{-3}$ | $2.54\ (1.81) \times 10^{-3}$ | $4.18\ (1.96) \times 10^{-3}$ | $3.71\ (2.30) \times 10^{-2}$ | $3.57\ (1.74) \times 10^{-2}$ | $2.07\ (1.19) \times 10^{-2}$ |





| | | | | | | |
|---|---|---|---|---|---|---|
| 2-Ethyltoluene (C$_9$H$_{12}$) | 6.89 (2.50) x 10$^{-3}$ | 2.70 (1.68) x 10$^{-3}$ | 4.63 (2.07) x 10$^{-3}$ | 3.76 (2.69) x 10$^{-2}$ | 3.39 (1.34) x 10$^{-2}$ | 2.10 (1.16) x 10$^{-2}$ |
| 1,3,5-Trimethylbenzene (C$_9$H$_{12}$) | 3.87 (1.71) x 10$^{-3}$ | 1.63 (1.22) x 10$^{-3}$ | 2.65 (1.43) x 10$^{-3}$ | 2.23 (1.60) x 10$^{-2}$ | 1.79 (0.83) x 10$^{-2}$ | 2.14 x 10$^{-2}$ (-) |
| 1,2,4-Trimethylbenzene (C$_9$H$_{12}$) | 1.04 (0.46) x 10$^{-2}$ | 4.25 (2.69) x 10$^{-3}$ | 7.52 (4.28) x 10$^{-3}$ | 6.23 (5.18) x 10$^{-2}$ | 3.91 (1.65) x 10$^{-2}$ | 1.74 (2.35) x 10$^{-2}$ |
| 1,2,3-Trimethylbenzene (C$_9$H$_{12}$) | 4.76 (2.59) x 10$^{-3}$ | 1.16 (0.81) x 10$^{-3}$ | 3.84 (2.69) x 10$^{-3}$ | 3.01 (3.16) x 10$^{-2}$ | 2.34 (0.43) x 10$^{-2}$ | 2.16 x 10$^{-2}$ (-) |
| Terpenes | | | | | | |
| alpha-Pinene (C$_{10}$H$_{16}$) | 8.30 (5.40) x 10$^{-4}$ | 5.38 (6.94) x 10$^{-4}$ | 7.82 (6.32) x 10$^{-4}$ | 2.26 (2.53) x 10$^{-3}$ | 0.35 (0.49) | 2.02 (2.33) x 10$^{-2}$ |
| beta-Pinene (C$_{10}$H$_{16}$) | 2.27 (1.49) x 10$^{-3}$ | 1.37 (0.91) x 10$^{-3}$ | 2.76 (3.15) x 10$^{-3}$ | 2.89 (3.56) x 10$^{-3}$ | 0.471 (-) | 4.67 x 10$^{-2}$ (-) |
| Oxygenates | | | | | | |
| Acetaldehyde (C$_2$H$_4$O) | 0.805 (0.279) | 0.334 (0.199) | 0.447 (0.119) | 1.70 (0.75) | 1.88 (1.63) | 0.541 (0.362) |
| Butanal (C$_4$H$_8$O) | 4.28 (1.50) x 10$^{-2}$ | 1.90 (1.29) x 10$^{-2}$ | 2.68 (1.05) x 10$^{-2}$ | 0.108 (0.047) | 5.40 (2.19) x 10$^{-2}$ | 8.28 (6.27) x 10$^{-3}$ |
| Acetone (C$_3$H$_6$O) | 0.705 (0.219) | 0.365 (0.226) | 0.416 (0.108) | 2.05 (0.52) | 1.63 (0.38) | 0.524 (0.256) |
| 2-Butanone (C$_4$H$_8$O) | 0.172 (0.057) | 8.00 (6.18) x 10$^{-2}$ | 0.103 (0.038) | 0.498 (0.151) | 0.262 (0.109) | 0.232 (0.286) |
| 2-Propenal (C$_3$H$_4$O) | 0.186 (0.060) | 0.127 (0.069) | 0.127 (0.059) | 0.295 (0.245) | nm | nm |
| MVK (C$_4$H$_6$O) | 0.129 (0.040) | 6.59 (4.56) x 10$^{-2}$ | 6.31 (2.76) x 10$^{-2}$ | 0.280 (0.147) | nm | nm |
| Furan (C$_4$H$_4$O) | 0.109 (0.041) | 5.98 (3.37) x 10$^{-2}$ | 6.81 (2.19) x 10$^{-2}$ | 0.379 (0.093) | 0.534 (0.209) | 0.241 (0.024) |
| 2-Methylfuran (C$_5$H$_6$O) | 0.117 (0.051) | 5.92 (4.77) x 10$^{-2}$ | 6.92 (2.83) x 10$^{-2}$ | 0.488 (0.227) | nm | nm |
| Furfural (C$_5$H$_4$O$_2$) | 8.55 (6.05) x 10$^{-2}$ | 4.28 (5.51) x 10$^{-2}$ | 8.22 (5.09) x 10$^{-2}$ | 0.316 (0.133) | nm | nm |
| Methanol (CH$_3$OH) | 2.09 (1.14) | 2.03 (2.01) | 1.18 (0.40) | 4.23 (3.40) | 2.38 (0.90) | 1.92 (0.61) |
| Ethanol (CH$_5$OH) | 4.08 (5.93) x 10$^{-2}$ | 2.18 (2.00) x 10$^{-2}$ | 5.63 (6.69) x 10$^{-2}$ | 7.62 (9.08) x 10$^{-2}$ | 0.563 (0.589) | 0.128 (0.017) |

*From Jayarathne et al. (2017), but part of same NAMaSTE study

nm indicates the species was not measured

(-) from Stockwell indicates that the measurement was not above background.





**Table 2. Emission factors (g VOC/ kg fuel C) for select compounds. The mean differences between dung/*angithi* and dung/*chulha* are shown and similarly for dung/*chulha* and brushwood/*chulha*. The significance between fuel or stove and EF is indicated with asterisks. Accompanying the mean differences is the average emission factor (g VOC/ kg fuel C) for dung cook fires and *chulha* cook fires, as well as the overall average for all performed cook fires.**

| Compound | Average EF for all cook fires (g/ kg fuel C) | *Angithi-chulha* average EF difference (g/ kg fuel C) | Average EF for dung fires (g/ kg fuel C) | Dung-brushwood average EF difference (g/ kg fuel C) | Average EF for *chulha* cook fires (g/ kg fuel C) |
|---|---|---|---|---|---|
| Ethane | 2.47 (2.16) | 4.18*** | 3.70 (2.43) | 1.19*** | 1.60 (0.744) |
| Propane | 0.827 (0.866) | 1.88*** | 1.32 (0.976) | 0.397*** | 0.448 (0.256) |
| n-Butane | 0.200 (0.236) | 0.52*** | 0.331 (0.271) | 0.0568*** | 0.097 (0.063) |
| Ethene | 4.17 (2.02) | N/A | 5.64 (1.32) | 4.05*** | 3.88 (2.07) |
| Propene | 2.24 (1.61) | 2.50*** | 3.38 (1.48) | 1.72*** | 1.63 (0.93) |
| 1-Butene | 0.473 (0.373) | 0.644*** | 0.718 (0.377) | 0.213*** | 0.327 (0.180) |
| Ethyne | 2.32 (1.41) | -2.46*** | 2.58 (1.63) | 2.21*** | 2.61 (1.37) |
| 1-Propye | 0.196 (0.108) | -0.129** | 0.244 (0.116) | 0.187*** | 0.204 (0.112) |
| 1-Butyne | $1.74 \times 10^{-2}$ ($7.74 \times 10^{-3}$) | -0.101*** | 0.219 (0.007) | 0.105*** | 0.017 (0.008) |

* denotes $p<0.05$, ** $p<0.01$, *** $p<0.001$