# Peer review of "Emissions from village cookstoves in Haryana, India and their potential impacts on air quality"

_Atmospheric Chemistry and Physics, 2018_

## Referee Comment (RC1) · Anonymous Referee #1 · 12 Jul 2018

Fleming et al. measured VOC and PM2.5 emissions from cook-stove fires in a rural village in India. Emissions were monitored from two stoves, angithi and chulha, and two fuel types, dung and brushwood. Emissions were generally higher for fuel-stove combinations that yielded the most smoldering combustion (lowest modified combustion efficiency). The authors have further assessed the impacts of emissions on OH reactivity, ozone formation, and SOA formation potential. Alkenes dominated the OH reactivity and ozone formation whereas aromatic hydrocarbons dominated the potential SOA. The work is generally well done and adds important information to the available emission inventories for real-world cook-stove fires. I recommend publication following consideration of the minor comments below.

Specific comments:

[Figure]

Pg 2, line 21: Please give a brief description of each stove type so that the reasons behind the differences in emissions are clearer.

Pg 3, line 1: Was the dung:brushwood ratio known for each mixed fire to accurately estimate fuel C content?

Pg 4, line 12: "The background filter mass was adjusted to match the flow rate of the sample filter by assuming the flow rate is proportional to the filter mass." This sentence is confusing. How would mass "match" a flow rate? Are the authors saying the sampled volume of the background filter was not always the same as that of the PM sample and therefore the background mass was scaled up or down according to the ratio of the sample volumes? Also, in Figure 1, the flow rate of the background filter is nominally the same as the PM filter, so were these marginal adjustments? Please rephrase this sentence.

Pg 7, line 12: In the comparison of EFs between this work and Stockwell et al., some consideration should be given to MCE. MCE could account for some of the EF differences for brushwood given the higher MCE of this study compared to Stockwell, although it is unlikely to explain the differences for dung. Is there sufficient information in Stockwell et al. (2016) to calculate EFs in g/kg C that could be overlaid on Figure 3 (or use g/kg in figure 3)?

Section 3.2: The overall discussion section would likely flow better if the MCE section is moved earlier, as most of the preceding discussion requires consideration of MCE.

Pg 7, line 28: In Figure 3b, the EFs from the chulha stove appear to follow a clear linear trend with MCE regardless of fuel type. In contrast, emissions from the angithi stove show no apparent trend with MCE. The benzene case is also similar. Could these "complicated" MCE relationships be due to differences between the two stoves? The authors should discuss the different MCE trends observed between the different stove types.

Pg 7, line 30: Alkyne emissions from the chulha stove clearly show increasing EF with decreasing MCE (negative slope). This contradicts the discussion on pg 6, line 34 that alkynes are predominantly emitted from flaming combustion, which would show a positive EF vs. MCE slope. Please reconcile those two points.

Pg 8, line 36: "aromatics make up on average roughly 95% of SOA precursors for all cook fires." This is not at all surprising considering that nearly all of the other compounds measured are either too light to contribute significantly to SOA production or don't have a reported SOAP value (Table S3). A simple disclaimer is warranted stating as such and that the contribution of aromatic hydrocarbons to SOAP here is likely an upper limit depending on the composition of the unmeasured fraction of VOCs in cookstove emissions. The authors may additionally want to compare to Stockwell et al. 2015 (Atmos. Chem. Phys., 15, 845-865), who measured laboratory cooking fires using PTR-TOFMS and observed many other compounds that could act as SOA precursors.

Pg 9, lines 7 and 12: Do these predicted ozone mixing ratios represent the excess ozone produced from only cooking fires? Clarify the text.

Section 4: The structure would make more sense if the paragraphs were swapped.

Technical corrections: Pg 2, line 14: typo in 'Alternatvely'

Pg 2, line 17: typo in "a simulated village houses".

Equation 1: It's redundant to use the summation sign and '+' signs.
* * *

---

## Referee Comment (RC2) · Anonymous Referee #2 · 9 Aug 2018

Fleming et al. report on gas-phase emissions from combinations of two cookstoves (angithi and chulha) and two fuel types (dung and brushwood) in a real world environment. They find that use of dung fuel and angithi cookstove results in higher gas-phase emissions, which is in line with lower observed modified combustion efficiencies. They use the gas-phase speciation to reflect on the potential of those emissions on ozone ($O_3$) and secondary organic (SOA) production.

The experimental and analysis methods of this study are robust and well executed and the findings reported agree very well with the data. The technical communication is of high quality and very easy to follow. I do not have any reservations with publication of this work in ACP subject to the authors responding to my comments below.

1. There is a wide diversity of stove and fuel types globally and the stove and fuel

[Figure]

types explored here are but a small fraction of those used in real world. So while the gas-phase speciation offers a detailed view of the emissions from these stove-fuel sources, how are the stove-fuel sources in this work representative of the stove-fuel combinations in India and globally? More specifically, what fraction of the gas-phase emissions from cookstoves come from the stove-fuels described in this work? And depending on the answers to the previous question, how can this speciation, if at all, be used to inform the gas-phase emissions speciation in large-scale atmospheric models?

2. Since multiple tests were done with each stove-fuel combination, were other important variables recorded and/or controlled during the test? For example, fuel moisture content, environmental conditions (e.g., temperature, relative humidity), fuel size and burn rate, cooking pots, meals cooked. According to the authors, are any of these variables important in explaining the variability? Citing relevant literature on the factors affecting cookstove emissions variability would be helpful.

3. Some more detail on the fuels and stoves for the less informed reader would be helpful. What animals was the dung from? Presumably cow? What is an Angithi stove? What is a chulha? How are these different? What is brushwood? Is the brushwood from a particular plant? The use of pictures could help.

4. In equations 1-3, how are $m_T$ and $m_{T,c}$ estimated/calculated?

5. Page 4, line 11: What is the limit of detection and limit of quantification for the filter measurement? The 0.75 $\mu$g blank seems quite low.

6. Page 5, line 1: I am not sure what the point is of normalizing the SOA production from a species to that of toluene? Why not report the SOA production in absolute values of g/kg-fuel when the presentation of results in Figure 2(c) is done in relative format anyways?

7. Why was the maximum incremental reactivity approach used to determine the ozone potential? If an alternative method was used (e.g., MOIR, EBIR), do the findings

change?

8. Can more details about the SOA formation be added? Were the SOA yields for low or high NOx conditions? What OA mass concentration as the absorbing mass was used to determine the SOA yield? Were they corrected for vapor wall losses? NOx and vapor wall loss corrections are species dependent (see Zhang et al., 2014) and may change the apportionment shown in Figure 2(c).

9. Page 7, lines 3-21: Are the brushwood results in this study comparable to the hardwood results from Stockwell et al. (2016)? If yes, why? It is unclear what the point of the comparison to the Stockwell study is since the manuscript only has a few sentences on this comparison. Is it just to show that the emissions in this study were lower than those in Stockwell et al. (2016). The explanations offered for the lower emissions was not satisfactory. Did the authors try to compare the emissions on a normalized basis with each other? Do they correlate?

10. Section 3.3 for SOA: Was there an estimate for emissions of total non-methane organic gases (NMOG)? What fraction did the speciated compounds account for of the total NMOG? Were any intermediate volatility and semi-volatile organic compounds targeted? What fraction of the NMOG was unspeciated? What implications does this unspeciated fraction that may include lower-volatility vapors have for SOA formation?

11. Section 4: Could atmospheric implications for SOA formation also be determined for this village similar to those for O3? How would the SOA formation compare to the primary PM2.5 emissions?

---

## Author Response (AR1)

*Comments by reviewer #1 are reproduced in the sans-serif font below. Our responses follow each comment in a blue, italicized, serif font. Text additions to the manuscript appear in the manuscript in red color. Deletions from the manuscript are described in the responses below.*

*In addition to the changes triggered by reviewer comments, other changes were made to improve readability. The significant additional additions or changes also show up in red color in the manuscript.*
* * *
Pg 2, line 21: Please give a brief description of each stove type so that the reasons behind the differences in emissions are clearer.

*The text was amended to clarify what the stoves are generally used for and the primary combustion type. Additions to the text are highlighted in red. The authors agree that these additions will assist the readers in making sense of the different emissions from the stoves that are described later.*

"This time, we measured emissions in field conditions from two traditional, locally-made cookstoves, the *chulha* and the *angithi*. The former is a primarily flaming stove with generally higher modified combustion efficiencies or concentration ratios of carbon dioxide to the sum of carbon dioxide and carbon monoxide (average dung-*chulha* 0.865), used to cook village meals. The *angithi* is largely smoldering with lower modified combustion efficiencies (average dung-*angithi* 0.819) and is primarily used for cooking animal fodder and simmering milk."

Additional text was added on page 3.

Pg 3, line 1: Was the dung:brushwood ratio known for each mixed fire to accurately estimate fuel C content?

*Yes it was. The following sentence was added to clarify this point.*

"When mixtures of dung and brushwood were used, both were individually weighed to accurately determine the carbon mass burned."

Pg 4, line 12: "The background filter mass was adjusted to match the flow rate of the sample filter by assuming the flow rate is proportional to the filter mass." This sentence is confusing. How would mass "match" a flow rate? Are the authors saying the sampled volume of the background filter was not always the same as that of the PM sample and therefore the background mass was scaled up or down according to the ratio of the sample volumes? Also, in Figure 1, the flow rate of the background filter is nominally the same as the PM filter, so were these marginal adjustments? Please rephrase this sentence.

*We agree this was confusing as written. We clarified that Teflon A (sample) and Teflon C (background) of Figure 1 were reserved for gravimetric analysis, whereas Teflon B was not used in this paper. With this it becomes clear that the background mass needs to be adjusted for flow rate, which is now described as matching sampling volumes in the text. We also explicitly state our assumption that the amount of PM collected on the filter is proportional to the sampling flow rate through the filter.*

Pg 7, line 12: In the comparison of EFs between this work and Stockwell et al., some consideration should be given to MCE. MCE could account for some of the EF differences for brushwood given the higher MCE of this study compared to Stockwell, although it is unlikely to explain the differences for

dung. Is there sufficient information in Stockwell et al. (2016) to calculate EFs in g/kg C that could be overlaid on Figure 3 (or use g/kg in figure 3)?

*Thank you for making this suggestion. We plotted their raw EF data in the supplemental information (Figure S2.2). There is more agreement in our datasets than it first appeared. This spurred a conversation with the authors of Stockwell et al., 2016 during which we learned that much of the discrepancy could come from adjusting their laboratory EFs to better match field observations. We also added text to explain this in the paper (page 7-8).*

Section 3.2: The overall discussion section would likely flow better if the MCE section is moved earlier, as most of the preceding discussion requires consideration of MCE.

*We believe that the emission factors are the most impactful contribution of this manuscript. Therefore, we decided to show first the emission factors in the chemical composition section. The statement that emissions generally have an inverse relationship with MCE is common knowledge, and does not warrant extensive discussion at the beginning of the paper. Rather, we focus on compounds/compound classes that have interesting relationships with MCE.*

Pg 7, line 28: In Figure 3b, the EFs from the chulha stove appear to follow a clear linear trend with MCE regardless of fuel type. In contrast, emissions from the angithi stove show no apparent trend with MCE. The benzene case is also similar. Could these "complicated" MCE relationships be due to differences between the two stoves? The authors should discuss the different MCE trends observed between the different stove types.

*We agree with this interpretation, and convey that the fuel-stove combination is important for determining emissions (not just MCE) in the second paragraph of the MCE section 3.2. A sentence was added to emphasize this point.*

"Knowledge of the cook fire MCE alone is not sufficient to determine the EF of ethane. Combustion specific to the fuel-stove combination is a significant factor in cook fire emissions."

Pg 7, line 30: Alkyne emissions from the chulha stove clearly show increasing EF with decreasing MCE (negative slope). This contradicts the discussion on pg 6, line 34 that alkynes are predominantly emitted from flaming combustion, which would show a positive EF vs. MCE slope. Please reconcile those two points.

*At lower MCEs there is a positive slope, this is largely encompassed by the dung-burning stoves. As there is more flaming combustion, we do in fact measure higher emissions of alkynes. However, it is well known that overall emissions are lower at higher MCEs. If one compares emission factors from ethane, alkenes, and alkynes in Figure 3, alkynes are highest at MCEs>0.95.*

Pg 8, line 36: "aromatics make up on average roughly 95% of SOA precursors for all cook fires." This is not at all surprising considering that nearly all of the other compounds measured are either too light to contribute significantly to SOA production or don't have a reported SOAP value (Table S3). A simple disclaimer is warranted stating as such and that the contribution of aromatic hydrocarbons to SOAP here is likely an upper limit depending on the composition of the unmeasured fraction of VOCs in cookstove emissions. The authors may additionally want to compare to Stockwell et al. 2015 (Atmos. Chem. Phys., 15, 845-865), who measured laboratory cooking fires using PTR-TOFMS and observed many other compounds that could act as SOA precursors.

*While Stockwell et al. 2015 did measure EFs from laboratory cook fires using PTR-TOF-MS, we chose to focus on Stockwell et al. 2016 because they use similar fuels and some of these measurements are in the*

*field. We do acknowledge that we did not quantify every SOA precursor which could add up to a significant amount of SOA. We clarified that the 95% comes from only species measured in this study on page 9.*

Pg 9, lines 7 and 12: Do these predicted ozone mixing ratios represent the excess ozone produced from only cooking fires? Clarify the text.

*This was clarified by adding phrases "due solely to cook stove use", "from cooking.", and "excess" in front of the predicted ozone levels.*

Section 4: The structure would make more sense if the paragraphs were swapped.

*We chose to keep the order of the paragraphs in the "Atmospheric implications and conclusions" section. To us, the first paragraph gives us an implication for the work, which is the quantification of ozone production in a village from cooking. Then the last paragraph holds take home conclusions for the readers, the order given in the subtitle.*

Technical corrections:

Pg 2, line 14: typo in 'Alternatvely'

*Thank you for catching this. It is now spelled correctly.*

Pg 2, line 17: typo in "a simulated village houses".

*The grammar was corrected.*

Equation 1: It's redundant to use the summation sign and '+' signs.

*Equation 1 was fixed.*

*Comments by reviewer #2 are reproduced in the sans-serif font below. Our responses follow each comment in a blue, italicized, serif font. Text additions to the manuscript appear in the manuscript in red color. Deletions from the manuscript are described in the responses below.*
* * *
1. There is a wide diversity of stove and fuel types globally and the stove and fuel types explored here are but a small fraction of those used in real world. So while the gas-phase speciation offers a detailed view of the emissions from these stove-fuel sources, how are the stove-fuel sources in this work representative of the stove-fuel combinations in India and globally? More specifically, what fraction of the gas-phase emissions from cookstoves come from the stove-fuels described in this work? And depending on the answers to the previous question, how can this speciation, if at all, be used to inform the gas-phase emissions speciation in large-scale atmospheric models?

*The objectives of the current paper are not to produce representative values for global emissions. Stockwell et al., 2016 and this work are the most comprehensive measurements to date in producing speciated gas phase emission factors. The stoves and fuels measured in the current study are predominant in the Indo-Gangetic plains. As a part of this collaborative project, the Seinfeld group at Caltech is modeling secondary organic aerosol formation using CMAQ to describe rural North India.*

2. Since multiple tests were done with each stove-fuel combination, were other important variables recorded and/or controlled during the test? For example, fuel moisture content, environmental conditions (e.g., temperature, relative humidity), fuel size and burn rate, cooking pots, meals cooked. According to the authors, are any of these variables important in explaining the variability? Citing relevant literature on the factors affecting cookstove emissions variability would be helpful.

*Moisture content of the fuels did not correlate with emission factors; in all cases the $r^2<0.3$. In one case, we did see a negative relationship between emission and moisture content (Benzene, dung-angithi, $r^2=0.718$). We did plot the emission factors as a function of fuel moisture content for a fraction of compounds as examples shown below. We found that meal cooked is not a significant variable (p>0.05). We added the recorded data for each cook fire to the supplemental information, including variable information and emission factors so the dataset is publically accessible. The following statements were added to make this clear to the reader.*

"Fuel moisture content, fuel mass burned, and meals cooked were noted for each cook fire, and can be found in the supporting information." *(Page 3)*

"There are many factors that may lead to variability in biomass burning emissions including pyrolysis temperature (Chen and Bond, 2010), fuel moisture content (Tihay-Felicelli et al., 2017), and the wind speed/direction (Surawski et al., 2015), among others. Relationships between emissions and fuel moisture content (Figure S1) or meal cooked were not found to be significant for any compounds (p< 0.05). This paper focuses on the relationships between emissions and fuel-stove combination." *(Page 6)*

[Figure]

3. Some more detail on the fuels and stoves for the less informed reader would be helpful. What animals was the dung from? Presumably cow? What is an Angithi stove? What is a chulha? How are these different? What is brushwood? Is the brushwood from a particular plant? The use of pictures could help.

*The following was added to the introduction, to explain the hypothesis that these stove will lead to different emissions.*

"The former is a primarily flaming stove with generally higher modified combustion efficiencies or concentration ratios of carbon dioxide to the sum of carbon dioxide and carbon monoxide (average dung-*chulha* 0.865), used to cook village meals. The *angithi* is largely smoldering with lower modified combustion efficiencies (average dung-*angithi* 0.819) and is primarily used for cooking animal fodder and simmering milk."

*The pictures of stoves and the kitchen are already published in Fleming et al., 2018, but the reader is referred to this publication for more extensive information about the materials used in the cook fires. The following sentences were added to the experimental methods section.*

"Animal fodder simmers upon smoldering dung in a clay bowl, referred to as an *angithi*. *Chulha* stoves are made from bricks and a covering of clay, and the availability of oxygen from the packing of biomass fuels results in primarily flaming combustion. The *chulha* is used to cook most meals for the family in this village. Buffalo and cow dung patties and brushwood, in the form of branches and twigs, were used in *chulha* stoves, and for the 13 mixed fuel cooking events dung and brushwood were combined in a ratio determined by the cook's preference."

4. In equations 1-3, how are mT and mT,c estimated/calculated?

*The language in red was added to clearly show how $m_T$ and $m_{T,C}$ are obtained.*

"Fuels were weighed before they were burned, and the dry mass was calculated based on moisture content measurements. The ash was weighed after the cooking event and subtracted from the dry mass of the fuel giving the net dry fuel burned for the cooking event, $m_T$. When mixtures of dung and brushwood were used, both were individually weighed to more accurately determine the carbon mass burned. The fraction of carbon in the fuel used to yield $m_{T,C}$ was taken to be 0.33 for buffalo dung and 0.45 for brushwood fuels based on Smith et al. 2000."

5. Page 4, line 11: What is the limit of detection and limit of quantification for the filter measurement? The 0.75 µg blank seems quite low.

*We determined the method detection limit to be 9.3 micrograms based off the standard deviation of the field blanks multiplied by 3. All masses for the sample and background filters are above this limit of detection.*

6. Page 5, line 1: I am not sure what the point is of normalizing the SOA production from a species to that of toluene? Why not report the SOA production in absolute values of g/kg-fuel when the presentation of results in Figure 2(c) is done in relative format anyways?

*The SOAP value for a particular VOC, as described by Derwent et al., 2010, is the change in SOA mass concentration when this VOC is introduced into the photochemical transport model, divided by the change in SOA mass concentration for when toluene is introduced. We use these SOAP values published in Derwent et al., 2010 to determine SOA forming potential for each quantified VOC. We are simply inheriting these data from the literature to estimate the relative magnitude of the effects, with the hope that more detailed modeling work would follow. The dataset is publicly available, so if one wished to generate absolute g SOA mass/ kg fuel burned by estimating the SOA yield for toluene in the plume, this is possible.*

7. Why was the maximum incremental reactivity approach used to determine the ozone potential? If an alternative method was used (e.g., MOIR, EBIR), do the findings change?

*Maximum incremental reactivity was chosen because a VOC limited regime gives us the highest sensitivity in ozone generation to cooking emissions. Carter 1994 concludes that the MIR scenario, since it is based on integrated ozone production rather than maximizing peak ozone levels (MOR scenario,) is generally less dependent on exact $NO_x$ inputs. This high-$NO_x$ scenario might be more realistic as well since we are focusing on smoke plumes. A sentence was added to the experimental section 2.7 summarizing this explanation for the use of the MIR scenario.*

8. Can more details about the SOA formation be added? Were the SOA yields for low or high NOx conditions? What OA mass concentration as the absorbing mass was used to determine the SOA yield? Were they corrected for vapor wall losses? NOx and vapor wall loss corrections are species dependent (see Zhang et al., 2014) and may change the apportionment shown in Figure 2(c).

*The photochemical trajectory model is outfitted with the Master Chemical Mechanism (MCM 3.1), which dictates gas-phase reactions and SOA formation. NO was initialized at 2 ppb, and $NO_2$ to 6 ppb. This is now described in section 2.5.*

9. Page 7, lines 3-21: Are the brushwood results in this study comparable to the hardwood results from Stockwell et al. (2016)? If yes, why? It is unclear what the point of the comparison to the Stockwell study is since the manuscript only has a few sentences on this comparison. Is it just to show that the emissions in this study were lower than those in Stockwell et al. (2016). The explanations offered for the lower emissions was not satisfactory. Did the authors try to compare the emissions on a normalized basis with each other? Do they correlate?

*Stockwell et al. 2016 and this paper are the most comprehensive gas-phase inventories to date in terms of cook fire emissions. It made sense to the authors to cross-compare the results. "Hardwood" and "brushwood" emissions appear to be more similar to each other than emissions from dung-based fuels. There are some real differences in emissions from cook fires between the studies, and we do our best to explain in the text these differences for the reader to compare. In S2.1 we plot the EFs in Stockwell et al. 2016 against those reported in our paper. In the review process, we've learned more about how the laboratory EFs were adjusted to field observations in Stockwell et al., 2016, and we believe these adjustments may have contributed to the discrepancy in the reported EFs. We've added Figure S2.2 where we plot the unadjusted emission factors in both studies, and there is an encouraging degree of agreement in the actually measured (unadjusted) values.*

10. Section 3.3 for SOA: Was there an estimate for emissions of total non-methane organic gases (NMOG)? What fraction did the speciated compounds account for of the total NMOG? Were any intermediate volatility and semi-volatile organic compounds targeted? What fraction of the NMOG was unspeciated? What implications does this unspeciated fraction that may include lower-volatility vapors have for SOA formation?

*We agree that there are semi and intermediate volatility compounds that were not measured in this study. Many of these will contribute to SOA and ozone formation. We did not attempt to quantify NMOG. This was not possible using our carbon balance method and an unknown dilution factor. We have added a "disclaimer" to the manuscript in the "SOA formation potential" section, printed below.*

"We would like to emphasize that the SOAP-weighted emissions are reflective of only the measured VOCs, and there are likely semi-volatile and intermediate volatility compounds that are not measured but also contribute to SOA formation."

11. Section 4: Could atmospheric implications for SOA formation also be determined for this village similar to those for O3? How would the SOA formation compare to the primary PM2.5 emissions?

*Unfortunately the SOA forming potentials generated using Derwent et al. 2010 only give relative amounts of SOA formation. From this analysis we concluded that a dung-chulha cook fire would produce almost 3 times more SOA than a brushwood-chulha cook fire. However, with this methodology we cannot compare the absolute magnitudes of primary and secondary emissions. Further work on addressing this exact question is in progress with the Seinfeld group at Caltech to model SOA formation in a simulated village.*

[revised manuscript text omitted]
$_8$) | 0.211 (0.073) | 9.48 (8.41) x 10$^{-2}$ | 0.116 (0.032) | 0.819 (0.157) | 0.457 (0.137) | 0.202 (0.140) |
| i-Butane (C$_4$H$_{10}$) | 1.73 (0.71) x 10$^{-2}$ | 4.60 (4.86) x 10$^{-3}$ | 9.51 (2.75) x 10$^{-3}$ | 7.27 (1.54) x 10$^{-2}$ | 0.215 (0.126) | 0.406 (0.478) |
| n-Butane (C$_4$H$_{10}$) | 4.71 (1.88) x 10$^{-2}$ | 1.57 (1.67) x 10$^{-2}$ | 2.68 (0.88) x 10$^{-2}$ | 0.215 (0.047) | 0.29 (0.09) | 1.11 (1.48) |
| n-Pentane (C$_5$H$_{12}$) | 2.01 (0.98) x 10$^{-2}$ | 4.44 (4.08) x 10$^{-3}$ | 9.12 (3.71) x 10$^{-3}$ | 6.80 (2.95) x 10$^{-2}$ | 0.190 (0.254) | 2.18 (1.73) x 10$^{-2}$ |
| n-Hexane (C$_6$H$_{14}$) | 1.03 (0.47) x 10$^{-2}$ | 1.96 (1.58) x 10$^{-3}$ | 5.31 (1.87) x 10$^{-3}$ | 4.93 (1.10) x 10$^{-2}$ | 0.291 (0.248) | 1.85 x 10$^{-2}$ (-) |
| n-Heptane (C$_7$H$_{16}$) | 7.21 (3.43) x 10$^{-3}$ | 9.23 (6.94) x 10$^{-4}$ | 3.92 (1.23) x 10$^{-3}$ | 3.17 (0.85) x 10$^{-2}$ | 0.114 (0.069) | 1.01 (1.35) x 10$^{-2}$ |
| 2-Methylpentane (C$_6$H$_{14}$) | 6.21 (2.81) x 10$^{-3}$ | 1.23 (0.99) x 10$^{-3}$ | 2.57 (1.61) x 10$^{-3}$ | 2.29 (1.67) x 10$^{-2}$ | 0.231 (0.192) | 9.93 (12.9) x 10$^{-3}$ |
| 3-Methylpentane (C$_6$H$_{14}$) | 3.71 (1.70) x 10$^{-3}$ | 1.21 (1.01) x 10$^{-3}$ | 1.57 (0.76) x 10$^{-3}$ | 7.54 (4.30) x 10$^{-3}$ | 0.155 (0.137) | 6.79 (6.63) x 10$^{-3}$ |
| Alkenes | | | | | | |
| Ethene (C$_2$H$_4$) | 1.86 (0.48) | 0.626 (0.284) | 1.13 (0.38) | 1.77 (0.35) | 4.23 (1.39) | 2.70 (1.17) |
| Propene (C$_3$H$_6$) | 0.807 (0.235) | 0.286 (0.202) | 0.417 (0.091) | 1.61 (0.33) | 1.47 (0.58) | 0.576 (0.195) |
| 1-Butene (C$_4$H$_8$) | 0.158 (0.047) | 6.32 (4.59) x 10$^{-2}$ | 8.38 (1.83) x 10$^{-2}$ | 0.366 (0.096) | 0.399 (0.331) | 0.726 (0.904) |
| i-Butene (C$_4$H$_8$) | 0.133 (0.057) | 3.46 (2.50) x 10$^{-2}$ | 6.40 (1.86) x 10$^{-2}$ | 0.353 (0.158) | 0.281 (0.091) | 0.846 (1.113) |
| trans-2-Butene (C$_4$H$_8$) | 4.45 (1.60) x 10$^{-2}$ | 2.00 (1.27) x 10$^{-2}$ | 2.38 (0.70) x 10$^{-2}$ | 0.151 (0.055) | 0.151 (0.010) | 6.78 (5.98) x 10$^{-2}$ |
| cis-2-Butene (C$_4$H$_8$) | 3.38 (1.19) x 10$^{-2}$ | 1.51 (0.95) x 10$^{-2}$ | 1.80 (0.52) x 10$^{-2}$ | 0.107 (0.047) | 0.102 (0.016) | 5.51 (4.76) x 10$^{-2}$ |
| 3-Methyl-1-butene (C$_5$H$_{10}$) | 1.46 (0.48) x 10$^{-2}$ | 5.74 (4.49) x 10$^{-3}$ | 7.30 (1.94) x 10$^{-3}$ | 3.82 (0.88) x 10$^{-2}$ | 5.58 (3.50) x 10$^{-2}$ | 7.43 (5.79) x 10$^{-3}$ |
| 2-Methyl-1-butene (C$_5$H$_{10}$) | 2.71 (1.28) x 10$^{-2}$ | 9.96 (10.9) x 10$^{-3}$ | 1.19 (0.42) x 10$^{-2}$ | 7.70 (3.99) x 10$^{-2}$ | nm | nm |
| 2-Methyl-2-butene (C$_5$H$_{10}$) | 2.51 (1.26) x 10$^{-2}$ | 6.40 (4.78) x 10$^{-3}$ | 1.10 (0.47) x 10$^{-2}$ | 9.17 (4.70) x 10$^{-2}$ | nm | nm |
| 1-Pentene (C$_5$H$_{10}$) | 4.17 (1.59) x 10$^{-2}$ | 9.65 (6.55) x 10$^{-3}$ | 2.13 (0.60) x 10$^{-2}$ | 0.122 (0.033) | 0.168 (0.086) | 1.43 (0.94) x 10$^{-2}$ |

| Compound | | | | | |
|---|---|---|---|---|---|
| trans-2-Pentene ($C_5H_{10}$) | 1.74 (0.65) x $10^{-2}$ | 8.89 (5.77) x $10^{-3}$ | 8.69 (2.22) x $10^{-3}$ | 5.14 (2.70) x $10^{-2}$ | 0.115 (0.035) | 1.05 (0.83) x $10^{-2}$ |
| cis-2-Pentene ($C_5H_{10}$) | 1.00 (0.36) x $10^{-2}$ | 5.55 (3.62) x $10^{-3}$ | 4.98 (1.26) x $10^{-3}$ | 2.50 (1.28) x $10^{-2}$ | 5.14 (0.76) x $10^{-2}$ | 8.69 x $10^{-3}$ (-) |
| 1-Hexene ($C_6H_{12}$) | 6.10 (2.46) x $10^{-2}$ | 1.26 (0.73) x $10^{-2}$ | 3.09 (0.91) x $10^{-2}$ | 0.167 (0.050) | nm | nm |
| 1,2-Propadiene ($C_3H_4$) | 3.76 (1.69) x $10^{-2}$ | 1.31 (0.62) x $10^{-2}$ | 2.32 (0.86) x $10^{-2}$ | 1.80 (0.923) x $10^{-2}$ | 7.15 (6.76) x $10^{-2}$ | 2.33 (1.07) x $10^{-2}$ |
| 1,2-Butadiene ($C_4H_6$) | 5.54 (1.68) x $10^{-3}$ | 2.82 (1.81) x $10^{-3}$ | 3.10 (1.06) x $10^{-3}$ | 4.33 (1.59) x $10^{-3}$ | nm | nm |
| 1,3-Butadiene ($C_4H_6$) | 0.203 (0.071) | 7.44 (3.99) x $10^{-2}$ | 0.108 (0.061) | 0.263 (0.082) | 0.409 (0.306) | 0.204 (0.144) |
| Isoprene ($C_5H_8$) | 8.94 (5.80) x $10^{-2}$ | 1.98 (1.48) x $10^{-2}$ | 3.03 (2.39) x $10^{-2}$ | 0.188 (0.143) | 0.325 (0.443) | 4.16 (2.23) x $10^{-2}$ |
| 1,3-Pentadiene ($C_5H_8$) | 1.96 (1.05) x $10^{-2}$ | 9.17 (4.79) x $10^{-3}$ | 9.39 (6.43) x $10^{-3}$ | 5.66 (2.94) x $10^{-2}$ | nm | nm |
| Alkynes | | | | | | |
| Ethyne | 1.13 (0.42) | 0.467 (0.160) | 0.890 (0.323) | 0.325 (0.238) | 0.593 (0.443) | 0.764 (0.363) |
| 1-Propyne | 9.42 (3.46) x $10^{-2}$ | 3.82 (1.76) x $10^{-2}$ | 5.99 (2.22) x $10^{-2}$ | 5.20 (2.83) x $10^{-2}$ | nm | nm |
| 1-Buten-3-yne ($C_4H_4$) | 5.04 (1.72) x $10^{-2}$ | 1.86 (0.90) x $10^{-2}$ | 3.46 (1.53) x $10^{-2}$ | 1.74 (1.26) x $10^{-2}$ | nm | nm |
| 1-Butyne ($C_4H_6$) | 7.72 (2.29) x $10^{-3}$ | 4.07 (2.24) x $10^{-3}$ | 4.48 (1.41) x $10^{-3}$ | 5.97 (1.93) x $10^{-3}$ | 2.29 (1.38) x $10^{-2}$ | 1.28 (0.47) x $10^{-2}$ |
| 2-Butyne ($C_4H_6$) | 4.31 (1.15) x $10^{-3}$ | 2.55 (1.44) x $10^{-3}$ | 2.47 (0.70) x $10^{-3}$ | 4.52 (1.40) x $10^{-3}$ | 1.86 (0.91) x $10^{-2}$ | 1.02 (0.66) x $10^{-2}$ |
| 1,3-Butadyne ($C_4H_2$) | 6.07 (2.66) x $10^{-3}$ | 2.71 (1.21) x $10^{-3}$ | 5.43 (2.01) x $10^{-3}$ | 1.53 (1.31) x $10^{-3}$ | nm | nm |
| Aromatics | | | | | | |
| Benzene ($C_6H_6$) | 1.03 (0.33) | 0.373 (0.149) | 0.723 (0.218) | 0.769 (0.175) | 1.96 (0.45) | 1.05 (0.19) |
| Toluene ($C_7H_8$) | 0.483 (0.273) | 0.221 (0.085) | 0.297 (0.077) | 0.860 (0.167) | 1.26 (0.05) | 0.241 (0.160) |
| Ethylbenzene ($C_8H_{10}$) | 3.41 (0.791) x $10^{-2}$ | 1.25 (1.20) x $10^{-2}$ | 1.97 (0.40) x $10^{-2}$ | 9.78 (1.66) x $10^{-2}$ | 0.366 (0.085) | 4.19 (4.25) x $10^{-2}$ |
| m/p-Xylene ($C_8H_{10}$) | 6.36 (1.26) x $10^{-2}$ | 2.78 (1.56) x $10^{-2}$ | 4.03 (0.98) x $10^{-2}$ | 0.148 (0.030) | 0.601 (0.294) | 9.57 (7.99) x $10^{-2}$ |
| o-Xylene ($C_8H_{10}$) | 2.38 (0.76) x $10^{-2}$ | 8.37 (5.78) x $10^{-3}$ | 1.44 (0.41) x $10^{-2}$ | 7.96 (1.91) x $10^{-2}$ | 0.228 (0.083) | 3.93 (4.31) x $10^{-2}$ |
| Styrene ($C_8H_8$) | 5.88 (1.58) x $10^{-2}$ | 2.28 (1.50) x $10^{-2}$ | 3.40 (1.90) x $10^{-2}$ | 8.63 (5.96) x $10^{-2}$ | 0.255 (0.091) | 8.71 (6.69) x $10^{-2}$ |
| i-Propylbenzene ($C_9H_{12}$) | 2.91 (0.77) x $10^{-3}$ | 1.20 (1.11) x $10^{-3}$ | 1.69 (0.45) x $10^{-3}$ | 9.30 (4.90) x $10^{-3}$ | 1.87 (1.40) x $10^{-2}$ | 1.70 (1.67) x $10^{-2}$ |
| n-Propylbenzene ($C_9H_{12}$) | 6.48 (2.59) x $10^{-3}$ | 1.84 (1.65) x $10^{-3}$ | 4.02 (1.59) x $10^{-3}$ | 3.95 (2.69) x $10^{-2}$ | 3.10 (1.45) x $10^{-2}$ | 1.78 (1.58) x $10^{-2}$ |
| 3-Ethyltoluene ($C_9H_{12}$) | 1.44 (0.48)x $10^{-2}$ | 5.46 (4.40) x $10^{-3}$ | 8.59 (3.26) x $10^{-3}$ | 7.14 (4.13) x $10^{-2}$ | 5.61 (2.38) x $10^{-2}$ | 2.62 (0.54) x $10^{-2}$ |

| | | | | | | |
|---|---|---|---|---|---|---|
| 4-Ethyltoluene ($C_9H_{12}$) | $6.35 (2.36) \times 10^{-3}$ | $2.54 (1.81) \times 10^{-3}$ | $4.18 (1.96) \times 10^{-3}$ | $3.71 (2.30) \times 10^{-2}$ | $3.57 (1.74) \times 10^{-2}$ | $2.07 (1.19) \times 10^{-2}$ |
| 2-Ethyltoluene ($C_9H_{12}$) | $6.89 (2.50) \times 10^{-3}$ | $2.70 (1.68) \times 10^{-3}$ | $4.63 (2.07) \times 10^{-3}$ | $3.76 (2.69) \times 10^{-2}$ | $3.39 (1.34) \times 10^{-2}$ | $2.10 (1.16) \times 10^{-2}$ |
| 1,3,5-Trimethylbenzene ($C_9H_{12}$) | $3.87 (1.71) \times 10^{-3}$ | $1.63 (1.22) \times 10^{-3}$ | $2.65 (1.43) \times 10^{-3}$ | $2.23 (1.60) \times 10^{-2}$ | $1.79 (0.83) \times 10^{-2}$ | $2.14 \times 10^{-2}$ (-) |
| 1,2,4-Trimethylbenzene ($C_9H_{12}$) | $1.04 (0.46) \times 10^{-2}$ | $4.25 (2.69) \times 10^{-3}$ | $7.52 (4.28) \times 10^{-3}$ | $6.23 (5.18) \times 10^{-2}$ | $3.91 (1.65) \times 10^{-2}$ | $1.74 (2.35) \times 10^{-2}$ |
| 1,2,3-Trimethylbenzene ($C_9H_{12}$) | $4.76 (2.59) \times 10^{-3}$ | $1.16 (0.81) \times 10^{-3}$ | $3.84 (2.69) \times 10^{-3}$ | $3.01 (3.16) \times 10^{-2}$ | $2.34 (0.43) \times 10^{-2}$ | $2.16 \times 10^{-2}$ (-) |
| Terpenes | | | | | | |
| alpha-Pinene ($C_{10}H_{16}$) | $8.30 (5.40) \times 10^{-4}$ | $5.38 (6.94) \times 10^{-4}$ | $7.82 (6.32) \times 10^{-4}$ | $2.26 (2.53) \times 10^{-3}$ | 0.35 (0.49) | $2.02 (2.33) \times 10^{-2}$ |
| beta-Pinene ($C_{10}H_{16}$) | $2.27 (1.49) \times 10^{-3}$ | $1.37 (0.91) \times 10^{-3}$ | $2.76 (3.15) \times 10^{-3}$ | $2.89 (3.56) \times 10^{-3}$ | 0.471 (-) | $4.67 \times 10^{-2}$ (-) |
| Oxygenates | | | | | | |
| Acetaldehyde ($C_2H_4O$) | 0.805 (0.279) | 0.334 (0.199) | 0.447 (0.119) | 1.70 (0.75) | 1.88 (1.63) | 0.541 (0.362) |
| Butanal ($C_4H_8O$) | $4.28 (1.50) \times 10^{-2}$ | $1.90 (1.29) \times 10^{-2}$ | $2.68 (1.05) \times 10^{-2}$ | 0.108 (0.047) | $5.40 (2.19) \times 10^{-2}$ | $8.28 (6.27) \times 10^{-3}$ |
| Acetone ($C_3H_6O$) | 0.705 (0.219) | 0.365 (0.226) | 0.416 (0.108) | 2.05 (0.52) | 1.63 (0.38) | 0.524 (0.256) |
| 2-Butanone ($C_4H_8O$) | 0.172 (0.057) | $8.00 (6.18) \times 10^{-2}$ | 0.103 (0.038) | 0.498 (0.151) | 0.262 (0.109) | 0.232 (0.286) |
| 2-Propenal ($C_3H_4O$) | 0.186 (0.060) | 0.127 (0.069) | 0.127 (0.059) | 0.295 (0.245) | nm | nm |
| MVK ($C_4H_6O$) | 0.129 (0.040) | $6.59 (4.56) \times 10^{-2}$ | $6.31 (2.76) \times 10^{-2}$ | 0.280 (0.147) | nm | nm |
| Furan ($C_4H_4O$) | 0.109 (0.041) | $5.98 (3.37) \times 10^{-2}$ | $6.81 (2.19) \times 10^{-2}$ | 0.379 (0.093) | 0.534 (0.209) | 0.241 (0.024) |
| 2-Methylfuran ($C_5H_6O$) | 0.117 (0.051) | $5.92 (4.77) \times 10^{-2}$ | $6.92 (2.83) \times 10^{-2}$ | 0.488 (0.227) | nm | nm |
| Furfural ($C_5H_4O_2$) | $8.55 (6.05) \times 10^{-2}$ | $4.28 (5.51) \times 10^{-2}$ | $8.22 (5.09) \times 10^{-2}$ | 0.316 (0.133) | nm | nm |
| Methanol ($CH_3OH$) | 2.09 (1.14) | 2.03 (2.01) | 1.18 (0.40) | 4.23 (3.40) | 2.38 (0.90) | 1.92 (0.61) |
| Ethanol ($CH_5OH$) | $4.08 (5.93) \times 10^{-2}$ | $2.18 (2.00) \times 10^{-2}$ | $5.63 (6.69) \times 10^{-2}$ | $7.62 (9.08) \times 10^{-2}$ | 0.563 (0.589) | 0.128 (0.017) |

*From Jayarathne et al. (2017), but part of same NAMaSTE study

nm indicates the species was not measured

(-) from Stockwell et al. (2016) indicates that the measurement was not above background.

**Table 2. Emission factors (g VOC/ kg fuel C) for select compounds. The mean differences between dung/*angithi* and dung/*chulha* are shown and similarly for dung/*chulha* and brushwood/*chulha*. The significance between fuel or stove and EF is indicated with asterisks. Accompanying the mean differences is the average emission factor (g VOC/ kg fuel C) for dung cook fires and *chulha* cook fires, as well as the overall average for all performed cook fires.**

| Compound | Average EF for all cook fires (g/ kg fuel C) | *Angithi-chulha* average EF difference (g/ kg fuel C) | Average EF for dung fires (g/ kg fuel C) | Dung-brushwood average EF difference (g/ kg fuel C) | Average EF for *chulha* cook fires (g/ kg fuel C) |
|---|---|---|---|---|---|
| Ethane | 2.47 (2.16) | 4.18[***] | 3.70 (2.43) | 1.19[***] | 1.60 (0.744) |
| Propane | 0.827 (0.866) | 1.88[***] | 1.32 (0.976) | 0.397[***] | 0.448 (0.256) |
| n-Butane | 0.200 (0.236) | 0.52[***] | 0.331 (0.271) | 0.0568[***] | 0.097 (0.063) |
| Ethene | 4.17 (2.02) | N/A | 5.64 (1.32) | 4.05[***] | 3.88 (2.07) |
| Propene | 2.24 (1.61) | 2.50[***] | 3.38 (1.48) | 1.72[***] | 1.63 (0.93) |
| 1-Butene | 0.473 (0.373) | 0.644[***] | 0.718 (0.377) | 0.213[***] | 0.327 (0.180) |
| Ethyne | 2.32 (1.41) | -2.46[***] | 2.58 (1.63) | 2.21[***] | 2.61 (1.37) |
| 1-Propye | 0.196 (0.108) | -0.129[**] | 0.244 (0.116) | 0.187[***] | 0.204 (0.112) |
| 1-Butyne | $1.74 \times 10^{-2}$ ($7.74 \times 10^{-3}$) | -0.101[***] | 0.219 (0.007) | 0.105[***] | 0.017 (0.008) |

[*] denotes p<0.05, [**] p<0.01, [***] p<0.001

**Supporting Information**

**Emissions from village cookstoves in Haryana, India and their potential impacts on air quality**

Lauren T. Fleming,[1] Robert Weltman,[2] Ankit Yadav,[3] Rufus D. Edwards,[2] Narendra K. Arora,[3] Ajay Pillarisetti,[4] Simone Meinardi,[1] Kirk R. Smith,[4] Donald R. Blake,[1] and Sergey A. Nizkorodov[1,*]

[1]Department of Chemistry, University of California, Irvine, CA 92697, USA
[2]Department of Epidemiology, University of California, Irvine, CA 92697, USA
[3]The Inclen Trust, Okhla Industrial Area, Phase-I, New Delhi-110020, India
[4]School of Public Health, University of California, Berkeley, CA 94720, USA

*Correspondence to:* Sergey Nizkorodov (nizkorod@uci.edu)

**Table S1. Average emission factors and standard deviation of PM$_{2.5}$ and gas-phase species (g kg$^{-1}$ dry fuel carbon) for dung-*chulha*, brushwood-*chulha*, mixed-*chulha*, and dung-*angithi* cook fires. Sample size (n) was n=12 for dung-*chulha*, n=14 for brushwood-*chulha*, n=13 for mixed-*chulha*, n=10 for dung-*angithi*. SOAP values, k$_{OH}$, and MIR values used to calculate predicted SOAP-weighted potentials, OH reactivity, and ozone-forming potential are included for the quantified species. SOAP values were taken from Derwent et al. (2010), k$_{OH}$ were found in the NIST Chemical Kinetics Database, and MIR values are found in Carter et al. 1994.**

[revised manuscript text omitted]

a) Dung

b) Brushwood

**Figure S2.2 Emission factors (g/ kg fuel C) plotted as a function of modified combustion efficiency for select species. Open circles indicate cooking events conducted with *angithi* stoves, whereas filled squares indicate *chulha* stoves. Color indicates fuel, either brushwood (blue), dung (red), or mixed (purple). Crosses indicate measurements from Stockwell et al. 2016.**

[Figure]

---

## Author Response (AR2)

Comments to the Author:

Dear Authors:

Thank you for your careful consideration of the referee comments. I accept this manuscript for publication after attention to the two minor points below.

1) Please consider adding the information regarding the prevalent use of these stove types in the region to the introduction. This information is provided in the response to Referee 2 comment 1 and would be beneficial in providing overall context to the reader.

2) Please add the limit of detection information provided in your response to Referee 2 comment 5 to the methods section.
* * *
New additions to the manuscript in response to the editorial comments are in green. Previous additions to the mansucript in response to the reviewer comments are still colored in red.

1) We added this information to the manuscript on page 2.

[revised manuscript text omitted]
$_8$) | 0.211 (0.073) | 9.48 (8.41) x 10$^{-2}$ | 0.116 (0.032) | 0.819 (0.157) | 0.457 (0.137) | 0.202 (0.140) |
| i-Butane (C$_4$H$_{10}$) | 1.73 (0.71) x 10$^{-2}$ | 4.60 (4.86) x 10$^{-3}$ | 9.51 (2.75) x 10$^{-3}$ | 7.27 (1.54) x 10$^{-2}$ | 0.215 (0.126) | 0.406 (0.478) |
| n-Butane (C$_4$H$_{10}$) | 4.71 (1.88) x 10$^{-2}$ | 1.57 (1.67) x 10$^{-2}$ | 2.68 (0.88) x 10$^{-2}$ | 0.215 (0.047) | 0.29 (0.09) | 1.11 (1.48) |
| n-Pentane (C$_5$H$_{12}$) | 2.01 (0.98) x 10$^{-2}$ | 4.44 (4.08) x 10$^{-3}$ | 9.12 (3.71) x 10$^{-3}$ | 6.80 (2.95) x 10$^{-2}$ | 0.190 (0.254) | 2.18 (1.73) x 10$^{-2}$ |
| n-Hexane (C$_6$H$_{14}$) | 1.03 (0.47) x 10$^{-2}$ | 1.96 (1.58) x 10$^{-3}$ | 5.31 (1.87) x 10$^{-3}$ | 4.93 (1.10) x 10$^{-2}$ | 0.291 (0.248) | 1.85 x 10$^{-2}$ (-) |
| n-Heptane (C$_7$H$_{16}$) | 7.21 (3.43) x 10$^{-3}$ | 9.23 (6.94) x 10$^{-4}$ | 3.92 (1.23) x 10$^{-3}$ | 3.17 (0.85) x 10$^{-2}$ | 0.114 (0.069) | 1.01 (1.35) x 10$^{-2}$ |
| 2-Methylpentane (C$_6$H$_{14}$) | 6.21 (2.81) x 10$^{-3}$ | 1.23 (0.99) x 10$^{-3}$ | 2.57 (1.61) x 10$^{-3}$ | 2.29 (1.67) x 10$^{-2}$ | 0.231 (0.192) | 9.93 (12.9) x 10$^{-3}$ |
| 3-Methylpentane (C$_6$H$_{14}$) | 3.71 (1.70) x 10$^{-3}$ | 1.21 (1.01) x 10$^{-3}$ | 1.57 (0.76) x 10$^{-3}$ | 7.54 (4.30) x 10$^{-3}$ | 0.155 (0.137) | 6.79 (6.63) x 10$^{-3}$ |
| **Alkenes** | | | | | | |
| Ethene (C$_2$H$_4$) | 1.86 (0.48) | 0.626 (0.284) | 1.13 (0.38) | 1.77 (0.35) | 4.23 (1.39) | 2.70 (1.17) |
| Propene (C$_3$H$_6$) | 0.807 (0.235) | 0.286 (0.202) | 0.417 (0.091) | 1.61 (0.33) | 1.47 (0.58) | 0.576 (0.195) |
| 1-Butene (C$_4$H$_8$) | 0.158 (0.047) | 6.32 (4.59) x 10$^{-2}$ | 8.38 (1.83) x 10$^{-2}$ | 0.366 (0.096) | 0.399 (0.331) | 0.726 (0.904) |
| i-Butene (C$_4$H$_8$) | 0.133 (0.057) | 3.46 (2.50) x 10$^{-2}$ | 6.40 (1.86) x 10$^{-2}$ | 0.353 (0.158) | 0.281 (0.091) | 0.846 (1.113) |
| trans-2-Butene (C$_4$H$_8$) | 4.45 (1.60) x 10$^{-2}$ | 2.00 (1.27) x 10$^{-2}$ | 2.38 (0.70) x 10$^{-2}$ | 0.151 (0.055) | 0.151 (0.010) | 6.78 (5.98) x 10$^{-2}$ |
| cis-2-Butene (C$_4$H$_8$) | 3.38 (1.19) x 10$^{-2}$ | 1.51 (0.95) x 10$^{-2}$ | 1.80 (0.52) x 10$^{-2}$ | 0.107 (0.047) | 0.102 (0.016) | 5.51 (4.76) x 10$^{-2}$ |
| 3-Methyl-1-butene (C$_5$H$_{10}$) | 1.46 (0.48) x 10$^{-2}$ | 5.74 (4.49) x 10$^{-3}$ | 7.30 (1.94) x 10$^{-3}$ | 3.82 (0.88) x 10$^{-2}$ | 5.58 (3.50) x 10$^{-2}$ | 7.43 (5.79) x 10$^{-3}$ |
| 2-Methyl-1-butene (C$_5$H$_{10}$) | 2.71 (1.28) x 10$^{-2}$ | 9.96 (10.9) x 10$^{-3}$ | 1.19 (0.42) x 10$^{-2}$ | 7.70 (3.99) x 10$^{-2}$ | nm | nm |
| 2-Methyl-2-butene (C$_5$H$_{10}$) | 2.51 (1.26) x 10$^{-2}$ | 6.40 (4.78) x 10$^{-3}$ | 1.10 (0.47) x 10$^{-2}$ | 9.17 (4.70) x 10$^{-2}$ | nm | nm |
| 1-Pentene (C$_5$H$_{10}$) | 4.17 (1.59) x 10$^{-2}$ | 9.65 (6.55) x 10$^{-3}$ | 2.13 (0.60) x 10$^{-2}$ | 0.122 (0.033) | 0.168 (0.086) | 1.43 (0.94) x 10$^{-2}$ |

[revised manuscript text omitted]

*From Jayarathne et al. (2017), but part of same NAMaSTE study

nm indicates the species was not measured

(-) from Stockwell et al. (2016) indicates that the measurement was not above background.

**Table 2. Emission factors (g VOC/ kg fuel C) for select compounds. The mean differences between dung/*angithi* and dung/*chulha* are shown and similarly for dung/*chulha* and brushwood/*chulha*. The significance between fuel or stove and EF is indicated with asterisks. Accompanying the mean differences is the average emission factor (g VOC/ kg fuel C) for dung cook fires and *chulha* cook fires, as well as the overall average for all performed cook fires.**

| Compound | Average EF for all cook fires (g/ kg fuel C) | *Angithi-chulha* average EF difference (g/ kg fuel C) | Average EF for dung fires (g/ kg fuel C) | Dung-brushwood average EF difference (g/ kg fuel C) | Average EF for *chulha* cook fires (g/ kg fuel C) |
|---|---|---|---|---|---|
| Ethane | 2.47 (2.16) | 4.18[***] | 3.70 (2.43) | 1.19[***] | 1.60 (0.744) |
| Propane | 0.827 (0.866) | 1.88[***] | 1.32 (0.976) | 0.397[***] | 0.448 (0.256) |
| n-Butane | 0.200 (0.236) | 0.52[***] | 0.331 (0.271) | 0.0568[***] | 0.097 (0.063) |
| Ethene | 4.17 (2.02) | N/A | 5.64 (1.32) | 4.05[***] | 3.88 (2.07) |
| Propene | 2.24 (1.61) | 2.50[***] | 3.38 (1.48) | 1.72[***] | 1.63 (0.93) |
| 1-Butene | 0.473 (0.373) | 0.644[***] | 0.718 (0.377) | 0.213[***] | 0.327 (0.180) |
| Ethyne | 2.32 (1.41) | -2.46[***] | 2.58 (1.63) | 2.21[***] | 2.61 (1.37) |
| 1-Propye | 0.196 (0.108) | -0.129[**] | 0.244 (0.116) | 0.187[***] | 0.204 (0.112) |
| 1-Butyne | $1.74 \times 10^{-2}$ ($7.74 \times 10^{-3}$) | -0.101[***] | 0.219 (0.007) | 0.105[***] | 0.017 (0.008) |

[*] denotes $p<0.05$, [**] $p<0.01$, [***] $p<0.001$